# Short-term outcomes of pubertal suppression in a selected cohort of 12 to 15 year old young people with persistent gender dysphoria in the UK

**Polly Carmichael[1]\*, Gary Butler[1,2,3], Una Masic[1], Tim J. Cole[3], Bianca L. De Stavola[3], Sarah Davidson[1], Elin M. Skageberg[1], Sophie Khadr[3], Russell M. Viner**[3]

**1** Gender Identity Development Service (GIDS), Tavistock and Portman NHS Foundation Trust, London, United Kingdom, **2** Paediatric Endocrine Service, University College London Hospitals NHS Foundation Trust, London, United Kingdom, **3** UCL Great Ormond Street Institute of Child Health, University College London, London, United Kingdom

\* PCarmichael@tavi-port.nhs.uk

## Abstract

### Background

In adolescents with severe and persistent gender dysphoria (GD), gonadotropin releasing hormone analogues (GnRHa) are used from early/middle puberty with the aim of delaying irreversible and unwanted pubertal body changes. Evidence of outcomes of pubertal suppression in GD is limited.

### Methods

We undertook an uncontrolled prospective observational study of GnRHa as monotherapy in 44 12–15 year olds with persistent and severe GD. Prespecified analyses were limited to key outcomes: bone mineral content (BMC) and bone mineral density (BMD); Child Behaviour CheckList (CBCL) total t-score; Youth Self-Report (YSR) total t-score; CBCL and YSR self-harm indices; at 12, 24 and 36 months. Semistructured interviews were conducted on GnRHa.

### Results

44 patients had data at 12 months follow-up, 24 at 24 months and 14 at 36 months. All had normal karyotype and endocrinology consistent with birth-registered sex. All achieved suppression of gonadotropins by 6 months. At the end of the study one ceased GnRHa and 43 (98%) elected to start cross-sex hormones.

There was no change from baseline in spine BMD at 12 months nor in hip BMD at 24 and 36 months, but at 24 months lumbar spine BMC and BMD were higher than at baseline (BMC +6.0 (95% CI: 4.0, 7.9); BMD +0.05 (0.03, 0.07)). There were no changes from baseline to 12 or 24 months in CBCL or YSR total t-scores or for CBCL or YSR self-harm indices, nor for CBCL total t-score or self-harm index at 36 months. Most participants reported

**Data Availability Statement:** The data underlying this study are available from the UK Data Service (DOI: 10.5255/UKDA-SN-854413).

**Funding:** The author(s) received no specific funding for this work.

**Competing interests:** The authors have declared that no competing interests exist.

positive or a mixture of positive and negative life changes on GnRHa. Anticipated adverse events were common.

## Conclusions

Overall patient experience of changes on GnRHa treatment was positive. We identified no changes in psychological function. Changes in BMD were consistent with suppression of growth. Larger and longer-term prospective studies using a range of designs are needed to more fully quantify the benefits and harms of pubertal suppression in GD.

## Introduction

Gender dysphoria (GD) describes the experience of incongruence between an individual's experienced gender and the sex they were assigned at birth. GD [1] in children and young people, also known as Gender Incongruence [2] and previously known as Gender Identity Disorder (GID), is associated with considerable distress or impairment in social, school or other important areas of functioning [3,4]. Interventions include psychosocial support, therapy and medical or surgical interventions to align the body with the identified gender [3,5]. Terminology in this field can be challenging [6]. Here we use birth-registered sex to refer to the sex assigned at birth by clinicians based upon external genitalia [6]. Gender identity refers to a young person's personal sense of their gender. We use the terms 'continuation' and 'discontinuation' to refer to GD across childhood and adolescence.

GD in adolescence is highly likely to continue into adult life where gender dysphoria persists after the onset of puberty [3]. Those with earlier onset or more intense GD and those in whom the development of secondary sexual characteristics in puberty is associated with increasing gender dysphoria or psychological distress are more likely to have persistent GD [3,7]. In adolescents with severe and persistent GD, international [8] and national [9–11] guidelines recommend the use of treatments to suppress the rise in sex hormones (oestradiol or testosterone) in young people during puberty. Gonadotropin releasing hormone analogues (GnRHa) are synthetic peptides that work by stimulating gonadotropin release in a tonic fashion which desensitises the gonadotropin receptors, resulting in reversible suppression of sex hormone production.

In GD, GnRHa can be used from the early/middle stages of puberty with the aim of delaying irreversible and unwanted pubertal body changes and giving young people the opportunity to explore their gender identity during a period when puberty is not advancing [3]. This period also allows clinicians more time to assess the stability of young people's gender identity [6]. Despite this treatment being given in mid-puberty it is also called early puberty suppression, where 'early' refers to earlier than the historic practice of suppression after completion of puberty.

Pubertal suppression is currently practised in the majority of international centres across Europe, the Americas and Australasia, as evidenced by a recently published survey of 25 international centres by the European Society of Paediatric Endocrinology (ESPE) [12]. Pubertal suppression with GnRHa as monotherapy is a time-limited strategy, due to the potential for side effects with long-term use. In the UK, for those commencing under age 15 years, use of GnRHa alone ceases after 16 years when young people face a decision to return to the sex hormones produced by their body or begin cross-sex hormones [5]. There are limited data on the outcomes of pubertal suppression in the treatment of young people with GD [3,13]. A recent

systematic review included data on the physical and mental health outcomes of pubertal suppression using GnRHa in over 500 young people [4]. Longer-term follow-up data on pubertal suppression in GD are limited to individuals from four cohorts [14–19].

In 2011 a study was begun to evaluate the proximal outcomes of mid-pubertal suppression using GnRHa in young people with persistent GD (see http://gids.nhs.uk/our-early-intervention-study). Use in the UK began after mid-pubertal suppression had been incorporated into international guidelines [20] and had become available in the USA [21,22], the Netherlands [15], Australia [23] and a number of European countries. The Gender Identity Development Service (GIDS) at the Tavistock and Portman NHS Foundation Trust, London, is a national service for children and young people with GD, drawing from England, Wales and Ireland. Mid-pubertal suppression was offered by the GIDS from 2011 initially only within an ethically approved uncontrolled observational research study with prospective data collection, where all participants received GnRHa. We anticipated that we would recruit 10–15 young people per year for 3 years and follow them up to the end of monotherapy with GnRHa. At the time, a randomised controlled study was not considered feasible due to very small numbers and inability to retain participants in the control arm, as the control treatment would have resulted in progression into near complete puberty and an increasing number of UK families were accessing mid-pubertal suppression internationally. Allocation blinding was also not considered feasible in young people using a product requiring monthly injections.

Here we describe the short-term outcomes of 44 young people with GD from this research cohort, recruited aged 12–15 years and followed to the end of GnRHa monotherapy after age 16 years. This paper describes their medical, psychological and social outcomes during the GnRHa treatment pathway up to the point of decisions about whether or not to undertake further physical treatment. The aims of the study as defined at inception in 2011 were:

1. To evaluate the benefits and risks for physical and mental health and wellbeing of mid-pubertal suppression in adolescents with GD

2. To add to the evidence base regarding the efficacy of GnRHa treatment for young people with GD

3. To evaluate continuation and discontinuation of GD and the continued wish for gender reassignment within this group.

## Methods

We undertook an uncontrolled prospective observational study of GnRHa monotherapy in a highly selected group of young people with persistent and severe GD.

### Participants

The cohort consisted of 44 sequentially eligible young people, aged 12 to 15 years, who were recruited between April 2011 and April 2014 and who commenced GnRHa treatment between June 2011 and April 2015. They were all recruited from patients referred to the GIDS.

Eligibility criteria were chosen to match those used for a Netherlands cohort [24], namely that the young person:
A. is aged 12–15 years
B. Psychological criteria

1. has been seen by the GIDS for at least 6 months and attended at least 4 interviews for assessment and therapeutic exploration of their gender identity development.

2. psychological stability sufficient to withstand the stresses of medical treatment for GID.

3. fulfils the following criteria relating to GID:

   a. Throughout childhood (defined as over 5 years) the adolescent has demonstrated an intense pattern of cross-gendered behaviours and cross-gender identity.

   b. The adolescent has gender dysphoria that is significantly increased with the onset of puberty. Following assessment the clinician(s) working with the young person deem that there is a high likelihood of the young person experiencing severe psychological distress consequent on experiencing full pubertal development before pubertal suppression is implemented.

4. The young person and their parents/guardians are actively requesting pubertal suppression.

5. is able to give informed consent.

   C. Physical/medical criteria

1. is in established puberty:

   - For birth-registered males Tanner (genital and pubic hair (PH)) stage 3 and above.

   - For birth-registered females Tanner (breast and PH) stage 2 and above.
     The rationale for the sex difference was that the pubertal growth spurt which early intervention aims to avoid occurs typically two years earlier in females (Tanner stage 2–3) than in males (Tanner stage 3–4), thus earlier intervention is required in females.

2. has normal endocrine function and karyotype consistent with birth registered sex.

   Note that the presence of mildly elevated androgens in birth registered females consistent with polycystic ovarian syndrome is not an exclusion criterion.
   Exclusion criteria:

1. Inability to participate with full investigatory protocol e.g. needle phobia, failure to attend for tests and scans.

2. Body mass index (BMI) <2nd centile for age and birth-registered sex [20].

3. Serious psychiatric conditions (e.g. psychosis, bipolar condition, anorexia nervosa, severe body-dysmorphic disorder unrelated to GD).

4. Inability to give informed consent according to the Fraser/Gillick guidelines.

5. Low spine or hip bone mineral density (BMD) on DXA scan: more than 2 SD below expected BMD for age and birth-registered sex. In exceptional circumstances a low BMD was acceptable if:

   i. it was felt to be clinically appropriate by the treating clinicians, who felt that on the balance of risks, pubertal suppression was justified despite the later risk of osteoporosis

   ii. the young person and parents understood the risks of GnRHa treatment for bone density (i.e. potential risks of later osteoporosis)

   iii. The young person and parents consented to more frequent monitoring of BMD (repeat DXA scans 6 months after starting GnRHa and yearly thereafter while on GnRHa) despite the small DXA radiation dose

 iv. The young person and parents consented to stopping treatment if raw BMD fell whilst on GnRHa.

## The treatment

The treatment under study was suppression of puberty using the GnRHa *triptorelin* together with psychosocial support and therapy, from study entry until the end of the GnRHa mono-therapy pathway at age 16 years or older. GnRHa monotherapy ceased when young people either started cross-sex hormones (and continued on GnRHa) or stopped GnRHa. Treatment duration was therefore from 1 to 4 or 5 years depending on age at study entry. Consenting young people were given triptorelin 3.75mg by intramuscular injection every 28 days during the treatment period. Two participants who found monthly injections difficult were moved to a ten-weekly preparation of 11.25mg of triptorelin. The aim of treatment was to suppress gonadotropins and sex hormones to near pre-pubertal levels [13]. Continued regular atten-dance for psychological support and therapy throughout the study was a precondition of GnRHa prescription. In addition local psychological services provided support for co-occuring difficulties for participants as required.

## Procedures and pathway

All young people and families attending the GIDS during the study period were provided with an information leaflet about research underway within the unit. Those wishing to find out more about the study discussed it with their GIDS clinicians and those deemed likely to be eli-gible were given detailed written study information. Those wanting to participate were invited to a medical clinic at UCLH for an initial discussion. At the first medical clinic, young people and families were seen by a senior paediatric endocrinology clinician together with a senior GIDS clinician, who discussed with the family the then current state of knowledge and ratio-nale for treatment, eligibility criteria and potential risks and benefits of participation. Risks included the anticipated side-effects of GnRHa treatment including symptoms resulting from the withdrawal of sex steroids (headaches, hot flushes), fatigue, loss of libido and low mood, the potential that treatment could influence the continuation of their GD and the potential for unknown risks. It was emphasised that young people needed to continue with both regular medical and psychosocial follow-up during the study and that treatment would cease if they did not comply with the treatment or monitoring requirements. A full medical history was elicited and the clinicians also reviewed a summary of the psychological history and assess-ment from the GIDS. In this visit information sheets were re-provided if families had lost them or forgotten details of the study. If young people and families remained interested in par-ticipation, medical investigations were organised and families were invited for a repeat discus-sion and a formal evaluation of eligibility at a second medical clinic visit approximately 3 months later. Families were asked to think about the issues raised in the meeting and to discuss with their GIDS clinicians if necessary, in order to discuss further at the second visit.

 At the second medical clinic visit, the same clinicians repeated the discussion of risks and benefits and explored understanding with the young person and family. A chaperoned medical examination was undertaken including pubertal assessment and the results of medical investi-gations were reviewed. Endocrine and GIDS clinicians jointly reviewed eligibility and offered participation in the study to those deemed eligible.

 The implications of treatment for fertility were discussed at the first and second medical vis-its and all young people were urged to consider storing gametes before starting GnRHa. Access to storage depended on regional availability within the NHS. Note that counselling on fertility

continued across the study, and clinicians periodically checked with young people who had decided against storage whether they wished to revisit their decision.

Informed consent was obtained in writing from both the young person and a parent or carer holding parental responsibility. The ability of the young person and parents to give informed consent was assessed jointly by the senior adolescent endocrine and GIDS clinicians, informed by written notes from the GIDS team. The consent forms were read with the young person and the parent by the clinicians to be sure they fully understood the information on the forms before signing.

48 young people and families attended the medical clinics for discussion of participation in the trial, of whom 44 wished to participate. Eight young people (7 birth assigned males) were not eligible for participation at the second medical visit as they were not yet sufficiently advanced in puberty. They were followed up every 3–6 months and entered the study subsequently when sufficiently advanced in puberty (median waiting time 7 months).

The date of signing the consent form was taken as the start of study treatment, although it frequently took one to three months for GnRHa treatment to start due to administrative requirements. Participants were followed up in the endocrine clinic, 3–6 monthly in the first 18 months and 12-monthly thereafter, till the end of the treatment pathway, defined as the date on or after the 16th birthday when a decision was made to either cease GnRHa or start cross-sex hormones. The final participant completed the pathway in February 2019.

## Outcomes

The following data were collected:

*A. Baseline explanatory variables*

1. Sex and gender: Young people were classified by their sex assigned at birth (birth-registered sex) and self-identified gender.

2. Ethnicity: Ethnicity was obtained from clinic records. For analysis, ethnicity was grouped as white, South Asian, black or mixed.

3. Puberty: Pubertal status at baseline was classified using information on genital/breast and pubic hair Tanner stages as appropriate. This was summarized into a single pubertal stage, with the breast/genital stage taking precedence if there was discrepancy between breast/genital and public hair stage.

4. Clinical data: These consisted of a) identification of normal phenotype on physical examination for birth-registered sex; b) venepuncture assessment of endocrinology (gonadotropins, prolactin, oestrogen or testosterone, adrenal androgens, thyroid function; and a short synachen test in birth-registered females only), karyotype, full blood count, renal and liver function, calcium and vitamin D; and c) imaging including wrist bone age and (in birth-registered females only) pelvic ultrasound scan. Medical assessment at baseline and follow-up was consistent with Endocrine Society guidelines [8,20].

*B. Study outcomes*

Study outcomes concerned domains including response to treatment, bone health, safety indicators and adverse events, psychological function; participant experience and satisfaction; and decisions regarding treatment following GnRHa. Outcome data were collected at routine clinic visits to GIDS or medical clinics at UCLH and timings therefore varied. For the purposes of these analyses, data for each participant were assigned to baseline (before treatment) and to the closest of the following outcome periods: 12, 24, 36 and 48 months on treatment. For safety and response to pubertal suppression outcomes, data were also examined at 6 months.

1. Response to pubertal suppression

Gonadotropins (LH, FSH), testosterone (in birth-registered males) and oestrogen (birth-registered females) were measured after venepuncture. Height, weight and blood pressure were recorded by trained clinic staff. BMI z-score for age and birth-registered sex was calculated [25]. Menarcheal status and presence/absence of menstrual periods was obtained by report from birth-registered females.

2. Bone health

Bone mineral content (BMC) and bone mineral density (BMD) in the lumbar (L1 to L4) spine and hip (total hip) were measured by dual energy X-ray absorptiometry (DEXA) scans using a Hologic Discovery QDR series model 010–1549 (Hologic Inc, Bedford, MA, USA). BMD z-scores for age and birth-registered sex appropriate to this machine were calculated [26]. BMD z-scores for spine and hip were further adjusted for height (height-adjusted z-scores) using published formulae [27].

3. Safety indicators and adverse events

Blood samples were collected by venepuncture for liver and renal function, full blood count, calcium and vitamin D, prolactin, adrenal androgens and thyroid function. Participants were routinely questioned about adverse events at medical clinic visits, including anticipated events such as headaches, hot flushes or fatigue plus any other unanticipated events.

4. Psychological function

Psychological outcomes included a clinical outcome routinely collected after GIDS appointments and a range of outcomes assessed using questionnaires. A standardised set of psychological questionnaires used in the GIDS clinic was completed at the time young people were deemed potentially eligible and referred to the medical clinic. Questionnaires were completed at home by the young person and parent between GIDS clinical meetings, and a research assistant followed up families to ensure their completion. Questionnaires were repeated approximately every 12 months on treatment.

i. General psychological functioning

The Child Behaviour Checklist (CBCL) (parent report) and Youth Self Report (YSR) (self-report) are general measures of psychological functioning and part of the Achenbach System of Empirically Based Assessment (ASEBA; www.aseba.org). The CBCL consists of 113 questions and is validated for children aged 6–18 years in international population samples [28]. The YSR consists of 112 questions and is validated in international populations of young people aged 11–18 years [29]. Questions in both are scored on a three-point Likert scale (0 = absent, 1 = occurs sometimes, 2 = occurs often), with the time frame for item responses being the past six months. Scoring for both instruments provides a total problems score, an internalizing problems score (items which assess anxious/depressed, withdrawn-depressed, and somatic complaints) and an externalizing score (focusing on rule-breaking and aggressive behaviours). Each questionnaire was scored with Assessment Data Manager Software using ASEBA standard norms and t-scores were generated based on reference data for birth-registered sex and broad age-ranges (here 12–18 years). Higher scores indicate greater morbidity. To account for normative change within our age-range, we used international reference data [29] to transform YSR raw scores into z-scores for year of age. As reference data from the UK were not available, reference data from both Australia and the Netherlands were used.

ii. Self-harm index

Self-harm actions and thoughts were assessed through two questions in each of the CBCL (parent report) and YSR (self-report): Item 18 (I deliberately try to hurt or kill myself) and Item 91 (I think about killing myself). Possible responses for each question were 0 = not true, 1 = somewhat or sometimes true, or 2 = very true or often true. We followed previous studies in calculating a self-harm index score to avoid multiple statistical comparisons across

correlated categorical-response variables. The index was calculated as the sum of the two items in each scale to create an index from 0 to 4 for each of the CBCL and YSR [30–32], a higher score indicating greater self-harm thoughts and behaviour.

iii. Health related quality of life (HRQoL)

This was assessed through separate young person and parent Kidscreen-52 questionnaires, each consisting of 52 items which assess HRQoL across ten dimensions: physical well-being; psychological well-being; moods and emotions; self-perception; autonomy; relations with parents and home life; social support and peers; school environment; social acceptance (bullying); and financial resources. All items use five-point Likert-style scales to assess either the frequency (never-seldom-sometimes-often-always) of certain behaviours/feelings or the intensity of an attitude (not at all–slightly-moderately-very-extremely). The measure was developed for young people aged 8–18 years, with the recall period of one week. The questionnaires provide scores in the form of continuous t-scores for the ten subscales derived from a multinational European sample [33]. Lower scores indicate lower HRQoL, i.e. greater morbidity.

iv. Body image

The Body Image Scale (BIS) is a self-report measure of 30 items used to assess body image satisfaction or dissatisfaction validated for age 12+. The instrument considers 30 body features which the respondent is asked to rate in terms of satisfaction on a five-point scale (1 = very satisfied, 2 = satisfied, 3 = neutral, 4 = dissatisfied, and 5 = very dissatisfied). The BIS provides a total score in the form of a continuous score for the total scale as well as for three subscales assessing primary sexual characteristics, secondary sexual characteristics and 'neutral' characteristics (i.e. non-sexual characteristics, e.g. nose) [34]. Higher scores represent higher degrees of body dissatisfaction.

v. Gender dysphoria

The Utrecht Gender Dysphoria Scale (UGDS) is a self-report measure used to assess the intensity of GD validated for age 12+. It comprises of 12 statements with agreement on a five-point scale (1 = agree completely, 2 = agree somewhat, 3 = neutral, 4 = disagree somewhat, and 5 = disagree completely). There are separate versions for birth-registered males and females. Items are summed to give a single total score, with higher scores indicating greater GD.

vi. Clinical outcomes

The Children's Global Assessment Scale (CGAS) is a rating of functioning in children and young people aged 6–17 years, extensively used as a routine clinical measure in child and adolescent mental health services in the UK. Treating clinicians assign young people a single score between 1 and 100, based on a clinician's assessment of a range of aspects related to a child's psychological and social functioning, with the time period being the previous month. Higher scores indicate better functioning, with categories ranging from 'extremely impaired' (1–10) to 'doing very well' (91–100) [35].

5. Participant experience and satisfaction with GnRHa

Young people were invited to participate in semi-structured qualitative interviews at 6–15 months and 15–24 months after starting GnRHa. Interviews were conducted in person or by telephone with a research assistant. If young people were unavailable, questions were posted to be completed and returned. The interview consisted of 12 questions related to changes young people had experienced in ten domains since starting on GnRHa: life overall, memory, focus, sense of direction, mood, energy levels, relationships with friends, relationships with family, gender role and sexuality. For each domain, young people were asked first about the general direction of change in that domain (whether changes were positive, neutral, negative or mixed positive and negative) and then asked for examples of changes experienced and why they assigned the chosen change rating. At the end of the interview two further questions were asked about change in any other experiences (i.e. allowing open ended responses) and whether

young people wished to continue on GnRHa treatment. Note there was no interview conducted before young people started GnRHa. Interviews were recorded in contemporaneous written notes by the researcher. The questionnaire is provided in the S1 Appendix.

6. Further treatment decisions

Decisions made at the end of the GnRHa pathway were recorded in terms of which if any further treatment for GD young people chose.

Note that other measures of gender dysphoria (Gender Identity Interview; Recalled Childhood Gender Identity Scale) were specified in our original protocol, however they were discontinued during the study as: a) they were historical instruments with poor construct validity and the binary references to male and female roles were challenging for some participants; and b) repeated questioning about gender dysphoria resulted in some distress to respondents. Our protocol had originally included the ASEBA Teacher Report Form (TRF), however we were unable to obtain data from teachers so this outcome was dropped. The Social Responsiveness Scale (SRS) was a baseline only assessment of autistic traits; these data will be analysed in the future.

## Analysis plan

Analyses were conducted according to the Statistical Analysis and Dissemination Plan, lodged with the ethics committee that approved the study before the analysis started (see S2 Appendix: Statistical Analysis Plan). The analysis plan was designed to report data on all outcomes but to minimise the likelihood of chance findings due to the large number of outcomes and small sample size. Sample sizes necessarily varied across follow-up as young people were recruited at different ages (12–15 years) but left the study soon after their 16[th] birthday. All 44 participants had data at 12 months follow-up. As participants necessarily left the study soon after their 16[th] birthday, numbers reduced after 12 months follow-up as participants could no longer remain in the study. Note this does not represent drop-out. There were 24 left at 24 months, 14 at 36 months and 4 at 48 months. In view of this, outcome reporting was restricted to change from baseline to 12, 24 and 36 months. We made no attempt to account for missing data due to the small sample size and the likelihood of the data missing not at random.

We restricted analyses to primarily descriptive statistics, with formal statistical testing of change across the study restricted to six pre-specified outcomes, i.e.:

1. Overall psychological functioning

    a. parent report: CBCL total t-score

    b. young person self-report: YSR total t-score

2. Self-harm index

    a. parent report: CBCL self-harm index

    b. young person self-report: YSR self-harm index

3. Bone health

    a. BMD and BMC for lumbar spine

    b. BMD and BMC for hip

Assessment of change was through paired t-tests for normally distributed data and the Wilcoxon matched-pairs sign-rank test for non-normal data. The number of formal statistical tests conducted in the study was 16; with overall significance at p = 0.05 and a Bonferroni correction, the appropriate threshold for statistical significance is about p = 0.003.

In our results and conclusions we refer to change in outcomes only for those that were formally tested. Reporting for other continuous outcomes was restricted to mean and 95% confidence intervals (95%CI) or median and interquartile range (IQR). For categorical outcomes, simple proportions were reported. We reported laboratory tests as normal or abnormal based upon laboratory reference data for age, with the exception of gonadotropins. We did not report data where the sample size was less than 8.

Analysis of potential predictors of outcome was confined a priori to two factors, birth-registered sex and pubertal stage at baseline. Three pre-specified continuous outcomes were examined at 12 months, namely:

1. BMD for lumbar spine

2. YSR total t-score

3. CGAS score

Associations were examined using linear regression of follow-up score on baseline score, adding each baseline factor separately to the model and considering the interaction of predictor with baseline score. All analyses were conducted using Stata 16 (Statacorp, College Station TX).

Responses to the semi-structured interview questionnaires were analysed simply for thematic content in terms of the direction and amount of change that young people experienced in each domain. This involved coding responses about experiences since starting GnRHa into categories; i.e. either positive/improving, negative/deteriorating, both positive and negative, no change or not known. The question on change in sexuality was coded as yes change, no change or not known. Wishes to continue with GnRHa were coded as yes, no or don't know.

To compare our findings with the literature, we drew upon recent reviews [3,4,6,13] and updated a recent review [4] from 1 June 2017 to 31 December 2019 using the same search terms in Medline (see S1 Appendix).

## Ethics

Ethical approval for the study was obtained from the National Research Ethics Service (NRES: reference 10/H0713/79) in February 2011. Study consent allowed the use of routinely collected clinical data (medical and psychological) as part of clinical treatment for the study. Study procedures including consent were reviewed by the UK Health Research Authority.

**Data sharing.** These are highly sensitive data from a small group of vulnerable young people treated in a single service and the risk of identification and disclosure is high. Research ethics permissions at the time the study was undertaken did not include permission to share data. After discussions with the Health Research Authority, UK, an anonymised dataset modified to remove sensitive data and minimise disclosure risk of personal information has been deposited with the UK Data Service.

## Results

Participants received psychosocial assessment and support within the GIDS before entering the study for a median of 2.0 years (IQR 1.4 to 3.2; range 0.7 to 6.6). The median time between first medical assessment at UCLH and starting treatment was 3.9 months (IQR 3.0 to 8.4; range 1.6 to 25.7). Median time in the study was 31 months (IQR 20 to 42, range 12 to 59).

Baseline characteristics of the participants by birth-registered sex are shown in Table 1. Median age at consent was 13.6 years (IQR 12.8 to 14.6, range 12.0 to 15.3). A total of 25 (57%) were birth-registered as male and 19 (43%) as female. At study entry, birth-registered males

**Table 1. Participant characteristics at baseline.**

| | | Total sample | Birth-registered sex | |
|---|---|---|---|---|
| | | n = 44 | male | female |
| | | | n = 25 | n = 19 |
| Age at consent (years) | Median (IQR) | 13.6 (12.8, 14.6) | 13.4 (12.7, 14.1) | 13.9 (13.5, 14.7) |
| Ethnic group n (%) | white | 39 (89) | 24 (96) | 15 (79) |
| | South Asian | 1 (2) | 1 (4) | 0 |
| | black | 2 (5) | 0 | 2 (11) |
| | Mixed ethnicity | 2 (5) | 0 | 2 (11) |
| Pubertal status n (%) | Stage 2 | 0 | 0 | 0 |
| | Stage 3 | 19 (43) | 17 (68) | 2 (10) |
| | Stage 4 | 16 (36) | 5 (20) | 11 (58) |
| | Stage 5 | 9 (21) | 3 (12) | 6 (32) |
| Menarcheal status n (%) | Premenarcheal | - | - | 4 (21) |
| | Post-menarcheal | - | - | 15 (79) |
| Time in study (months) | Median (IQR) | 31 (20, 42) | 37 (24, 43) | 29 (17, 36) |
| Age at end of pathway (years) | Median (IQR) | 16.1 (16.0, 16.4) | 16.1 (16.0, 16.5) | 16.1 (16.0, 16.3) |

At baseline, all participants had normal endocrinology, karyotype, imaging and clinical phenotype on physical examination for birth-registered sex and normal full blood count and liver and renal function. No participants had evidence of disorders of sexual differentiation. Eight participants (18%) had vitamin D insufficiency at baseline and were given vitamin D supplements.

were predominantly in stage 3 puberty (68%) whilst birth-registered females were predominantly in stages 4 (58%) or 5 (32%) with 79% (15/19) post-menarcheal. 89% of participants were of white ethnicity. Birth-registered females were on average 6 months older than birth-registered males at study entry.

## Response to treatment

All participants achieved adequate suppression of gonadotropins and sex hormones by 6 months (mean LH 0.5IU/L; mean FSH 1.4IU/L) and maintained it throughout the study (see Table 2). Liver function, basic haematology and biochemistry were normal in all participants at 3–6 months. All post-menarcheal birth-registered females reported amenorrhoea in the 3 months after starting GnRHa treatment and remained so throughout treatment. No participants reported progression in pubertal development. Height and weight were normal at baseline. Height growth continued through the study but more slowly than expected for age, thus

**Table 2. Growth and gonadotropin levels at baseline, 12, 24 and 36 months.**

| Growth | | Baseline | | 12 months | | 24 months | | 36 months | |
|---|---|---|---|---|---|---|---|---|---|
| | | n | Mean (95% CI) | n | Mean (95% CI) | n | Mean (95% CI) | n | Mean (95% CI) |
| Height | z-score | 44 | 0.4 (0.1, 0.7) | 44 | 0.2 (-0.1, 0.4) | 24 | 0.0 (-0.4, 0.4) | 14 | 0.0 (-0.5, 0.5) |
| Weight | z-score | 44 | 0.8 (0.4, 1.3) | 44 | 0.8 (0.3, 1.3) | 24 | 0.6 (-0.1 1.3) | 14 | 1.0 (0.1, 1.9) |
| BMI | z-score | 44 | 0.7 (0.2, 1.1) | 44 | 0.7 (0.2, 1.2) | 24 | 0.6 (-0.1, 1.3) | 14 | 1.1 (0.3, 1.9) |
| **Gonadotropins** | | | | | | | | | |
| LH | IU/L | 42* | 4.2 (2.8, 5.6) | 44 | 0.60 (0.42, 0.68) | 17 | 0.40 (0.22, 0.60) | 7 | 0.30 (0.14, 0.46) |
| FSH | IU/L | 42* | 3.9 (3.2, 4.5) | 44 | 1.3 (1.0, 1.7) | 17 | 1.0 (0.6, 1.5) | 7 | 1.4 (0.7, 2.2) |

*In two participants data recorded as normal at baseline were not available.

height z-score fell over time (Table 2). Weight and BMI z-scores were stable from baseline to 24 months but increased at 36 months.

Three participants had brief periods off GnRHa prior to their 16th birthday. In one, treatment was withdrawn by clinicians due to non-attendance at clinics and restarted 4 months later. Another requested a period off GnRHa to think further about treatment in view of other things happening in their life; they restarted 4 months later. A third, birth-registered male, stopped GnRHa for 9 months to attempt to store sperm, contrary to their earlier decision not to, and restarted afterwards.

Median age at the end of the GnRHa pathway was 16.1 years (Table 1). A quarter of participants made their decision more than six months later, either because they wished to delay due to school exams or other events or because clinicians felt they were not yet ready to make the decision. One young person decided to stop GnRHa and not start cross-sex hormones, due to continued uncertainty and some concerns about side-effects of cross-sex hormones. The remaining 43 (98%) elected to start cross-sex hormones.

**Bone mineral density.** BMD was available on 44 participants at baseline, 43 at 12 months, 24 at 24 months and 12 at 36 months (Table 3). Numbers were lower for hip than for spine as some hip scans were not done for technical reasons. The table shows mean values at baseline and 12, 24 and 36 months, along with mean baseline values corresponding to the paired samples at each time point. There was no change from baseline in spine or hip at 12 months nor in hip at 24 and 36 months, but at 24 months lumbar spine BMC and BMD were higher than at baseline, as was lumbar BMC at 36 months. Lumbar and hip BMD age-adjusted z-scores were in the normal range at baseline but point-estimates fell at 12 and 24 months but not at 36 months. Point-estimates for height-adjusted z-scores for lumbar and hip BMD also fell at 12 and 24 months but not at 36 months.

**Psychological outcomes.** For the standardised questionnaires, baseline assessments were conducted at a median of 0.5 (IQR 0.4, 0.8) years before starting treatment, and were available for all 44 participants by self-report and 43 by parental report. Data on the CBCL, YSR, Kidscreen-52, BIS and CGAS were normally distributed whilst those for UGDS and the CBCL and YSR self-harm indices were skewed.

The first psychological follow-up was at a median of 13 (IQR 12, 14) months after start of treatment, with ASEBA data available for 41 participants (parent and self-report). ASEBA data at 24 months (median 25 (21, 28)) were available on 20 young people by parent report and 15 by self-report, and at 36 months (median 36 (29, 39)) on 11 by parent report and 6 by self-report.

Formal testing was undertaken only for key ASEBA outcomes (Table 4). For the CBCL total t-scores, there was no change from baseline to 12, 24 or 36 months. Similarly for the YSR total t-score, there was no change from baseline to 12 or 24 months; YSR data at 36 months (n = 6) were not analysed. There were no significant changes in parent-report CBCL self-harm index scores from baseline to 12, 24 or 36 months, nor for self-report YSR self-harm index scores.

Other psychological outcomes are described in Table 5. Point-estimates of scores on the Kidscreen-52, BIS, UGDS and CGAS showed little change over time."

The pre-specified outcomes of BMD at lumbar spine, YSR total t-score and CGAS score at 12 months, adjusted separately for birth-registered sex and baseline pubertal status, along with the baseline level of the outcome, are shown in Table 6. None of the outcomes were associated with birth-registered sex or pubertal status, and there were no important interactions.

**Participant experience, satisfaction and side effects.** 41 participants completed interviews at 6–15 months (median 9) and 29 at 15–24 months (median 21); 3 missed both. Fig 1 shows proportions with positive or negative changes for life overall, mood and friendships, with summary data for all questions shown in S1 Appendix (S1 and S2 Tables).

**Table 3. Bone mineral density outcomes at baseline, 12, 24 and 36 months.**

| | | | | 12 months | | | | 24 months | | | |
|---|---|---|---|---|---|---|---|---|---|---|---|
| | | | Baseline | Baseline for those followed up | Follow-up | Change | p | Baseline for those followed up | Follow-up | Change | p |
| | | n | Mean (95% CI) | n | Mean (95% CI) | Mean (95% CI) | Mean (95% CI) | | n | Mean (95% CI) | Mean (95% CI) | Mean (95% CI) | |
| Lumbar | BMC | 44 | 39.5 (35.9, 43.1) | 42 | 39.6 (35.8, 43.4) | 41.2 (38.2, 44.2) | 1.6 (0.2, 3.1) | 0.03 | 24 | 34.1 (30.3, 37.9) | 40.1 (36.7, 43.5) | 6.0 (4.0, 7.9) | <0.0001 |
| | BMD | 44 | 0.76 (0.71, 0.80) | 43 | 0.76 (0.71, 0.80) | 0.77 (0.72, 0.81) | 0.01 (-0.00, 0.03) | 0.17 | 24 | 0.68 (0.63, 0.74) | 0.73 (0.68, 0.78) | 0.05 (0.03, 0.07) | 0.0001 |
| Hip | BMC | 43 | 25.2 (23.2, 27.1) | 39 | 25.5 (23.4, 27.6) | 26.1 (24.4, 27.9) | 0.7 (-0.2, 1.5) | 0.13 | 22 | 23.9 (21.2, 26.6) | 26.3 (24.1, 28.6) | 2.4 (0.7, 4.1) | 0.008 |
| | BMD | 43 | 0.80 (0.75, 0.86) | 39 | 0.81 (0.75, 0.87) | 0.82 (0.78, 0.86) | 0.01 (-0.02, 0.05) | 0.6 | 22 | 0.76 (0.68, 0.85) | 0.79 (0.74, 0.84) | 0.03 (-0.04, 0.10) | 0.4 |
| BMD z-scores | Spine | 44 | -0.3 (-0.7, 0.0) | 43 | -0.3 (-0.7, 0.1) | -1.0 (-1.3, -0.7) | | | 24 | -0.5 (-1.1, 0.0) | -1.5 (-2.1, -0.8) | | |
| | HAZ spine | 44 | -0.5 (-0.8, -0.1) | 43 | -0.4 (-0.8, -0.1) | -1.0 (-1.3, -0.6) | | | 24 | -0.7 (-1.2, -0.1) | -1.3 (-1.9, -0.7) | | |
| | Hip | 43 | -0.5 (-0.9, -0.1) | 39 | -0.5 (-0.9, -0.1) | -1.0 (-1.3, -0.6) | | | 21 | -0.5 (-1.1, 0.1) | -1.4 (-2.0, -0.9) | | |
| | HAZ hip | 43 | -0.7 (-1.0, -0.3) | 39 | -0.6 (-1.0, -0.2) | -0.9 (-1.3, -0.5) | | | 21 | -0.5 (-1.1, 0.1) | -1.2 (-1.7, -0.6) | | |

| | | | | 36 months | | | | | | | | |
|---|---|---|---|---|---|---|---|---|---|---|---|---|
| | | | | Baseline for those followed up | Follow-up | Change | p | | | | | |
| | | | | n | Mean (95% CI) | Mean (95% CI) | Mean (95% CI) | | | | | | |
| Lumbar | BMC | | | 12 | 37.05 (31.0, 43.1) | 42.4 (37.4, 47.4) | 5.3 (2.8, 7.8) | 0.0007 | | | | |
| | BMD | | | 12 | 0.72 (0.65, 0.80) | 0.76 (0.70, 0.82) | 0.03 (.00, 0.07) | 0.05 | | | | |
| Hip | BMC | | | 12 | 26.1 (22.1, 30.0) | 26.8 (21.2, 32.3) | 0.7 (-3.8, 5.2) | 0.7 | | | | |
| | BMD | | | 12 | (0.82, 0.73, 0.91) | 0.81 (0.74, 0.88) | -0.009 (-0.05, 0.03) | 0.6 | | | | |
| BMD z-scores | Spine | | | 12 | -0.2 (-1.0, 0.6) | -1.5 (-2.2, -0.8) | | | | | | |
| | HAZ spine | | | 12 | -0.4 (-1.2, 0.3) | -1.3 (-2.2, -0.5) | | | | | | |
| | Hip | | | 12 | -0.3 (-1.3, 0.6) | -1.1 (-1.8, -0.5) | | | | | | |
| | HAZ hip | | | 12 | -0.5 (-1.5, 0.5) | -1.0 (-1.8, -0.2) | | | | | | |

BMD: bone mineral density; BMC bone mineral content; HAZ height adjusted z-score.

BMD z-scores were not formally tested–see Methods.

Most participants reported positive or a mix of positive-negative changes in their life at both time points. At 6–15 months 46% reported only positive changes, including feeling happier, relieved, less facial hair or stopping periods. A further 37% reported both positive and negative changes such as feeling happier but also experiencing hot flushes and headaches. In addition 12% reported overall negative changes namely hot flushes, tiredness, and feeling more emotional, while 5% reported no change. At 15–24 months, 55% reported solely positive changes such as feeling happier, no longer experiencing side effects and feeling more

**Table 4. ASEBA outcomes at baseline, 12, 24 and 36 months.**

| | | Baseline | | 12 months | | | | 24 months | | | |
|---|---|---|---|---|---|---|---|---|---|---|---|
| | | | | Baseline for those followed up | Follow-up | Change | p | Baseline for those followed up | Follow-up | Change | p |
| | | n | mean (95% CI) | n | mean (95% CI) | mean (95% CI) | mean (95% CI) | | n | mean (95% CI) | mean (95% CI) | mean (95% CI) | |
| Parent report CBCL | Total problems t-score | 43 | 61.6(58.4, 64.7) | 41 | 61.5(58.2, 64.7) | 61.8(58.4, 65.1) | 0.3(-2.0, 2.6) | 0.8 | 20 | 61.2(56.5, 65.8) | 60.2(54.6, 65.8) | -1.0(-4.0, 2.1) | 0.5 |
| | Externalising problems t-score | 43 | 55.8(52.4, 59.3) | 41 | 55.7(52.1, 59.3) | 55.4(51.8, 59.0) | | | 20 | 55.4(49.9, 60.9) | 55.2(48.9, 61.5) | | |
| | Internalising problems t-score | 43 | 62.1(58.7, 65.5) | 41 | 61.8(58.3, 65.3) | 62.9(59.5, 66.3) | | | 20 | 60.4(55.7, 65.1) | 60.1(54.6, 65.6) | | |
| Self-report YSR | Total problems t-score | 44 | 57.9(55.0, 60.8) | 41 | 57.6(54.5, 60.6) | 58.4(54.6, 62.2) | 0.8(-3.1, 4.8) | 0.7 | 15 | 55.1(50.9, 59.2) | 56.5(50.6, 62.5) | 1.5(-3.4, 6.3) | 0.5 |
| | Total problems z-score (ref: Netherlands) | 44 | 1.01(0.67, 1.36) | 41 | 0.97(0.62, 1.33) | 0.99(0.55, 1.42) | | | 15 | 0.66(0.17, 1.15) | 0.65(-0.05, 1.36) | | |
| | Total problems z-score (ref: Australia) | 44 | 0.72(0.37, 1.06) | 41 | 0.68(0.32, 1.03) | 0.68(0.24, 1.12) | | | 15 | 0.39(-0.11, 0.89) | 0.37(-0.32, 1.07) | | |
| | Externalising problems t-score | 44 | 52.3(49.2, 55.5) | 41 | 52.3(49.2, 55.4) | 52.5(48.7, 56.3) | | | 15 | 53.1(48.5, 57.6) | 52.3(45.3, 59.4) | | |
| | Internalising problems t-score | 44 | 58.0(54.9, 61.2) | 41 | 57.7(54.3, 61.0) | 60.1(55.9, 64.3) | | | 15 | 53.9(49.9, 58.0) | 55.9(50.8, 61.1) | | |
| **Self-harm scores** | | | | | | | | | | | | |
| Parent report CBCL | Median (IQR) | 43 | 0(0, 1) | 40 | 0(0, 1) | 0(0, 1) | | 0.3 | 20 | 0(0, 1) | 0(0, 1) | | >0.9 |
| Self-report YSR | Median (IQR) | 43 | 0(0, 1) | 39 | 0(0, 1) | 0(0, 2) | | 0.4 | 15 | 0(0, 0) | 0(0, 0) | | 0.3 |

| | | 36 months | | | |
|---|---|---|---|---|---|
| | | Baseline for those followed up | Follow-up | Change | **p** |
| | | n | mean (95% CI) | mean (95% CI) | mean (95% CI) | |
| Parent report CBCL | Total problems t-score | 11 | 62.4(55.1, 69.6) | 61.1(52.3, 69.9) | -1.3(-6.6, 4.0) | 0.6 |
| | Externalising problems t-score | 11 | 56.8(48.0, 65.6) | 56.2(48.3, 64.1) | | |
| | Internalising problems t-score | 11 | 60.4(53.5, 67.2) | 62.5(53.6, 71.5) | | |
| **Self-harm scores** | | | | | | |
| Parent report CBCL | Median (IQR) | 11 | 0(0, 1) | 0(0, 1) | | 0.8 |

comfortable with puberty suspended. A further 17% reported both positive and negative changes including less body hair but continued growth in height, or having clearer skin but also experiencing more hunger, weight gain and tiredness. 17% reported largely negative changes such as mood swings, tiredness and hot flushes whilst 10% reported no change.

Reports of change in mood were mixed. At 6–15 months, the majority reported mood to be improved (49%), mixed changes (such as both feeling happier but experiencing some mood swings; 15%) or no change (7%), however 24% reported negative changes in mood such as

**Table 5. Other psychological outcomes at baseline, 12, 24 and 36 months.**

| | | Baseline | | 12 months | | 24 months | | 36 months | |
|---|---|---|---|---|---|---|---|---|---|
| | | n | mean (95% CI) | n | mean (95% CI) | n | mean (95% CI) | n | mean (95% CI) |
| **Kidscreen-52 HRQOL** | | | | | | | | | |
| Parent report CBCL t-scores | Physical wellbeing | 42 | 44.9(41.4, 48.5) | 36 | 40.4(37.5, 43.3) | 14 | 40.5(36.8, 44.2) | | |
| | Psychological Wellbeing | 41 | 39.8(36.7, 42.8) | 36 | 39.0(35.4, 42.6) | 14 | 42.4(36.9, 48) | | |
| | Moods and Emotions | 41 | 40.6(37.6, 43.6) | 36 | 41.2(37.3, 45.1) | 14 | 42.5(36.3, 48.7) | | |
| | Self-perception | 42 | 34.6(32.6, 36.5) | 36 | 34.8(32.0, 37.5) | 14 | 34.8(31.3, 38.2) | | |
| | Autonomy | 42 | 46.2(43.2, 49.2) | 36 | 48.2(45.0, 51.4) | 14 | 46.7(41, 52.4) | | |
| | Parent relations and home life | 42 | 48.1(44.5, 51.6) | 35 | 46.7(42.9, 50.5) | 14 | 49.5(44.1, 54.9) | | |
| | Social support and peers | 39 | 48.0(44.7, 51.4) | 36 | 51.9(48.4, 55.3) | 13 | 51.4(45.6, 57.2) | | |
| | School environment | 42 | 38.2(35.0, 41.4) | 35 | 39.4(35.3, 43.4) | 13 | 43.7(36, 51.3) | | |
| | Social acceptance | 39 | 44.7(40.7, 48.7) | 32 | 42.3(38.1, 46.4) | 13 | 43.5(35.9, 51.2) | | |
| | Financial resources | 42 | 37.9(33.9, 41.9) | 36 | 35.8(31.5, 40.2) | 14 | 36.3(26.4, 46.3) | | |
| Self-report t-scores | Physical wellbeing | 42 | 45.1(41.8, 48.5) | 36 | 41.5(38.0, 45.0) | 13 | 43.9(38.9, 48.9) | | |
| | Psychological Wellbeing | 42 | 43.0(39.6, 46.4) | 36 | 41.1(37.0, 45.2) | 14 | 51(45.8, 56.2) | | |
| | Moods and Emotions | 42 | 46.3(42.7, 49.9) | 36 | 43.9(40.4, 47.3) | 14 | 50.1(45.5, 54.7) | | |
| | Self-perception | 42 | 38.8(36.7, 40.9) | 36 | 37.9(35.1, 40.6) | 14 | 43.1(39.9, 46.2) | | |
| | Autonomy | 42 | 46.6(43.6, 49.6) | 36 | 46.7(42.9, 50.5) | 13 | 51.9(47.4, 56.4) | | |
| | Parent relations and home life | 42 | 49.7(46.2, 53.2) | 36 | 48.7(45.2, 52.3) | 14 | 58.4(53.3, 63.5) | | |
| | Social support and peers | 37 | 45.6(42.5, 48.7) | 35 | 48.1(44.6, 51.6) | 14 | 49.7(44.3,55.1) | | |
| | School environment | 41 | 45.9(42.3, 49.4) | 36 | 44.7(39.7, 49.7) | 14 | 49(43.6, 54.3) | | |
| | Social acceptance | 41 | 47.4(43.5, 51.3) | 33 | 45.5(40.9, 50.1) | 13 | 53.6(46.3, 60.8) | | |
| | Financial resources | 42 | 42.2(38.1, 46.3) | 34 | 43.2(38.2, 48.1) | 14 | 46.3(39.1, 53.5) | | |
| **Body image scale** | Overall score | 42 | 3.1(2.8, 3.3) | 40 | 3.2(3.0, 3.4) | 16 | 3(2.7, 3.2) | 8 | 3.1(2.4, 3.7) |
| | Primary characteristics score | 42 | 4.5(4.2, 4.7) | 39 | 4.3(4.2, 4.5) | 16 | 4.5(4.3, 4.7) | 8 | 4.2(3.9, 4.5) |
| | Secondary characteristics score | 41 | 2.9(2.6, 3.1) | 40 | 3(2.8, 3.3) | 16 | 2.9(2.5, 3.2) | 8 | 2.9(2, 3.8) |
| | Neutral characteristics score | 42 | 2.5(2.203, 2.707) | 40 | 2.7(2.5, 3.0) | - | - | | |
| **Utrecht Gender dysphoria score** | Median (IQR) | 41 | 4.8(4.6, 5.0) | 40 | 4.7(4.6, 5.0) | 18 | 4.7(4.3, 5.0) | | |
| **Clinical outcome** | | | | | | | | | |
| CGAS global score | Mean (95% CI) | 42 | 62.9(59.6, 66.2) | 35 | 64.1(59.9, 68.3) | 18 | 65.7(59.6, 71.8) | 12 | 66.0(58.1, 73.9) |

Note: Change in outcomes in this Table were not formally tested.

**Table 6. Associations between birth-registered sex and baseline pubertal status and outcomes at 12 months.**

| | | Outcomes at 12 months adjusted for baseline | | | | | | | |
|---|---|---|---|---|---|---|---|---|---|
| | | BMD at lumbar spine | | | YSR total t-score | | | GCAS score | |
| | | n | Coefficient (95% CI) | p | n | Coefficient (95% CI) | p | n | Coefficient (95% CI) | p |
| Birth-registered sex | | | | | | | | | |
| Main effect (baseline value of outcome) | | 43 | 0.86 (0.75, 0.97) | <0.0001 | 41 | 0.43 (0.05, 0.82) | 0.03 | 33 | 0.74 (0.42, 1.06) | <0.0001 |
| Birth-registered sex | Male (ref) | | 0 | | | 0 | | | 0 | |
| | Female | | -0.02 (-0.05, 0.01) | 0.2 | | 2.1 (-5.2, 9.4) | 0.6 | | -3.2 (-10.0, 3.5) | 0.3 |
| Pubertal status | | | | | | | | | |
| Main effect (baseline value of outcome) | | 43 | 0.85 (0.72, 0.97) | <0.0001 | 41 | 0.43 (0.01, 0.84) | 0.04 | 33 | 0.69 (0.37, 1.00) | <0.0001 |
| Pubertal stage at baseline | 3 | | 0.008 (-0.03, 0.04) | 0.7 | | 0.2 (-8.3, 8.7) | 0.9 | | 1.6 (-5.5, 8.8) | 0.6 |
| | 4 (ref) | | 0 | | | 0 | | | 0 | |
| | 5 | | -0.009 (-0.05, 0.03) | 0.7 | | 0.4 (-9.9, 10.8) | 0.9 | | -7.9 (-17.6, 1.8) | 0.11 |

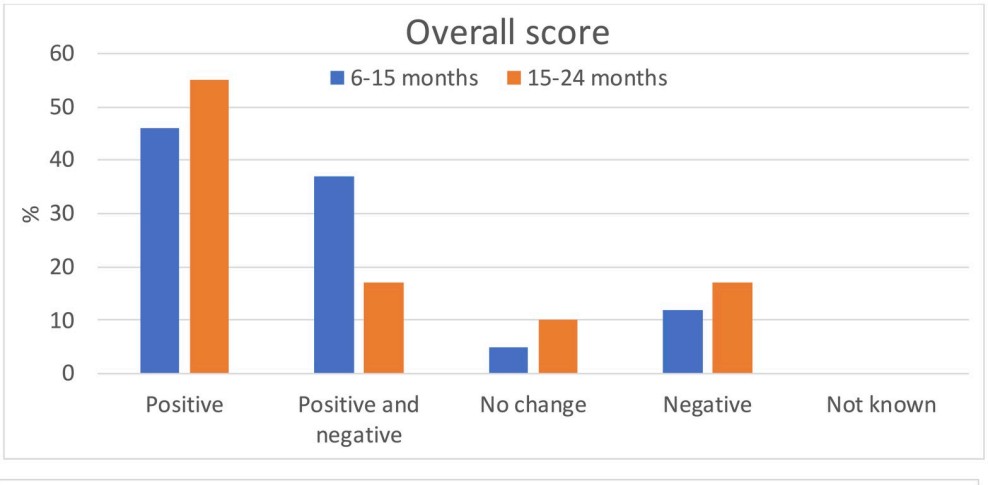

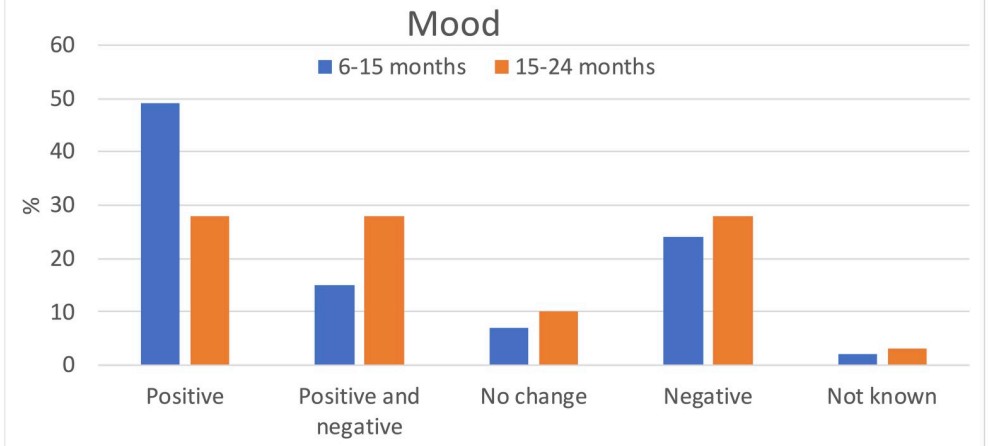

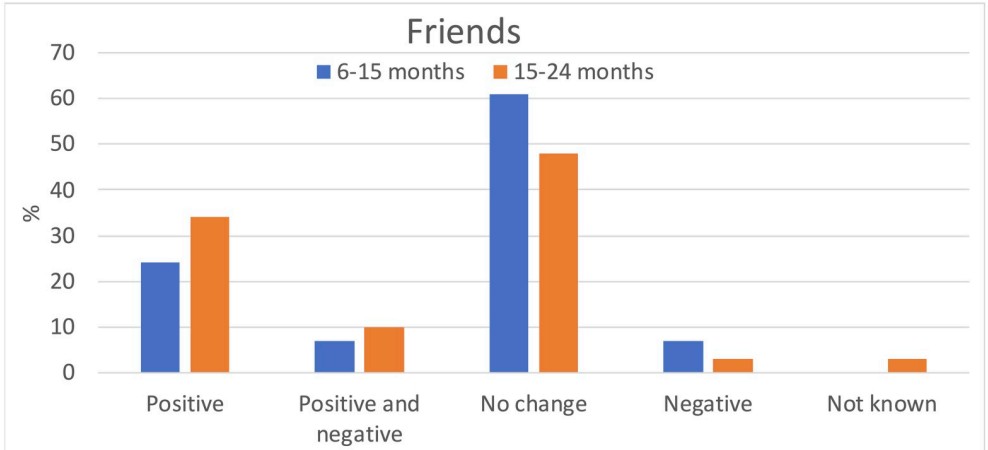

**Fig 1. Ratings of change in life overall, mood and friendships at 6–15 months (n = 41) and 15–24 months (n = 29).**

experiencing more mood swings or feeling low. Findings at 15–24 months were similar. The most common negative change was reduced energy levels, reported by 29% at 6-15m and 38% at 15-24m.

Young people's reports of change in family and peer relationships were predominantly positive or neutral at both time points. Positive changes included feeling closer to the family,

**Table 7. Adverse events reported across the study.**

| Participants | 0-6m | 7-12m | 13-24m | 25+m |
|---|---|---|---|---|
| | n = 44 | n = 44 | n = 36 | n = 24 |
| | *n (%)* | *n (%)* | *n (%)* | *n (%)* |
| Mild headaches or hot flushes | 11 (25%) | 10 (23%) | 8 (22%) | 4 (17%) |
| Moderate or severe headaches and hot flushes | 2 (5%) | 4 (9%) | 1 (3%) | 0 |
| Fatigue—mild | 2 (5%) | 3 (7%) | 3 (8%) | 1 (4%) |
| Fatigue–moderate or severe | 0 | 0 | 0 | 0 |
| Mood swings | 1 (2%) | 0 | 0 | 0 |
| Weight gain | 1 (2%) | 0 | 1 (3%) | 0 |
| Sleep problems | 1 (2%) | 0 | 1 (3%) | 0 |
| Other events | 0 | 0 | 0 | 0 |
| Total events recorded* | 18 | 17 | 14 | 5 |

* individuals may have more than 1 event.

feeling more accepted and having fewer arguments. Those reporting both positive and negative change reported feeling closer to some family members but not others. At 6–15 months, negative family changes were largely from family members not accepting their trans status or having more arguments. But by 15–24 months only one young person reported this. Improved relationships with peers related to feeling more sociable or confident and widening their circle of friends; negative changes related to bullying or disagreements at school. Again, at 15–24 months only one young person reported negative change, related to feelings of not trusting friends.

At 6–15 months, changes in gender role were reported by 66% as positive, including feeling more feminine/masculine, living in their preferred gender identity in more (or all) areas of life and feeling more secure in their gender identity, with no negative change reported. At 15–24 months, most reported no change although 41% reported positive changes including experimenting more with physical appearance and changing their details on legal documents.

All young people affirmed at each interview that they wished to continue with GnRHa treatment. Note that this was also the case when asked routinely at medical clinics (excepting those who briefly ceased GnRHa as noted above).

**Adverse events.** Adverse events are shown in Table 7. All adverse events were minor and anticipated, i.e. they were previously described in study participant information and/or noted in the triptorelin medication package inserts. Anticipated adverse events were common in the first two years, particularly mild headaches or hot flushes which were reported in 25% at 0-6m, 23% at 7-12m and 22% at 13-24m. Moderate or severe headaches and/or hot flushes were uncommon. Birth-registered females with distressing headaches or hot flushes were offered 'add-back' oestrogen therapy, and two accepted treatment briefly with very small doses of oestradiol, which was effective in reducing symptoms. Mild fatigue was reported by 5–8% over the first two years and no participants reported moderate or severe fatigue. Sleep problems, mood swings and weight gain were reported by very small numbers and in each case symptoms were mild. Adverse events were less common after 12 months of treatment.

## Discussion

We report the short and medium-term outcomes of a prospective cohort of 44 young people with persistent and severe GD treated with GnRHa resulting in pubertal suppression from mid-puberty for 1–4 years. Young people were considered for recruitment after lengthy

assessment, spending an average of 2 years and up to 6 years within the GIDS psychological service before being referred to the endocrine clinic for assessment to enter the study. Medical assessment found no endocrine abnormalities at baseline. GnRHa treatment started in the majority of participants in later stages of puberty, with 57% in puberty stages 4 and 5 and 79% of birth-registered females being post-menarcheal. After starting GnRHa all quickly achieved and maintained suppression of pubertal hormones and none experienced pubertal progression. At the end of the study, 43 (98%) chose to start cross-sex hormones whilst one young person chose to stop GnRHa and continue with puberty consistent with their birth-registered sex.

As anticipated, pubertal suppression reduced growth that was dependent on puberty hormones, i.e. height and BMD. Height growth continued for those not yet at final height, but more slowly than for their peers so height z-score fell. Similarly for bone strength, BMD and BMC increased in the lumbar spine indicating greater bone strength, but more slowly than in peers so BMD z-score fell. These anticipated changes had been discussed with all participants before recruitment to the study. Young people experienced little change in mean weight or BMI z-score in the first two years. The rise in weight and BMI z-score at 36 months may represent a trend towards greater adiposity in those on GnRHa for a prolonged period, or reflect a higher baseline in this group.

Information on side-effects was available through routine reporting in medical clinics and in the participant experience interviews. Anticipated side effects of treatment were common, particularly mild symptoms directly related to suppression of sex hormones. Severe symptoms were uncommon. Fatigue or low energy was reported rarely in medical clinic assessments but frequently at interview (38% at 15-24m). The relationship of symptoms such as headaches, fatigue and sleep disturbance to GnRHa treatment is unclear as they are all very common in early adolescence [36,37], although a conservative perspective would regard them as side-effects of treatment.

Young people experienced little change in psychological functioning across the study. We found no differences between baseline and later outcomes for overall psychological distress as rated by parents and young people, nor for self-harm. Outcomes that were not formally tested also showed little change.

Participant experience of treatment as reported in interviews was positive for the majority, particularly relating to feeling happier, feeling more comfortable, better relationships with family and peers and positive changes in gender role. Smaller numbers reported having mixed positive and negative changes. A minority (12% at 6–15 months and 17% at 15–24 months) reported only negative changes, which were largely related to anticipated side effects. None wanted to stop treatment due to side effects or negative changes. We are not aware of comparative patient experience data from other cohorts.

The median age at consent in our study was very similar to that in the earliest published outcome study of mid-pubertal suppression using GnRHa treatment in Dutch young people (13.6 years) [24]. Similarly to this Dutch cohort, all but one of our participants elected to start cross-sex hormones after completing the GnRHa pathway. However they spent an average of 31 months on GnRHa compared with 23 months in the Dutch cohort [24]. In our study, the successful suppression of puberty and cessation of menses with GnRHa, the impact on height growth [4,16,38] and BMD [4,16] and the normality of liver and renal function through treatment were each consistent with previous reports [4,16].

Our findings that BMD increased over time in the lumbar spine but more slowly than in same age peers, resulting in a fall in z-score, are similar to others [4,14,39,40]. The fall in height-adjusted BMD z-score was consistent with but larger than the fall in height z-score. We found that birth-registered sex and pubertal status at baseline were not associated with later BMD. There is evidence that accretion of bone mass resumes and that BMD increases with the

start of cross-sex hormone therapy [4,14,39,41]. Future research needs to examine longer-term change in BMD in young people treated with mid-pubertal suppression.

We reported a range of adverse events previously described to be associated with pubertal suppression [42], with the exception of mild sleep disturbance although this is a known association with triptorelin use. As anticipated, the withdrawal of sex hormones produces symptoms such as headaches and lack of energy, although in the great majority (11 of 13 at 0–6 months; 10 of 14 at 7–12 months; 8 of 9 at 13–24 months) the symptoms were minor. Symptoms diminished over time as has previously been noted [4], and no young people chose to cease treatment due to the side-effects.

Our finding that 1 participant ceased pubertal suppression and did not commence cross-sex hormones is somewhat similar to the experience of one US cohort and a second Dutch cohort; Kuper et al. described that 2 of approximately 57 young people aged 10–15 years who commenced pubertal suppression treatment stopped this treatment without commencing cross-sex hormones [17]. Brik et al. reported that in a cohort of 137 young people who began GnRHa between 10 and 18 years and were followed until eligible to commence cross-sex hormones, 5 (3.6%) ceased treatment and did not later commence cross-sex hormones [19].

Three longitudinal studies from the Netherlands and the USA have examined psychological function over time in cohorts of young people treated with GnRHa and then cross-sex hormones [17,18,24], although the two US cohorts were of limited size. Our study adopted the same psychological outcome measures as the Dutch cohort, to facilitate comparison [24]. Mean baseline YSR scores in our cohort were similar to those previously reported in 141 young people aged 12–18 years from the London GIDS [43], and baseline CBCL and YSR scores were close to those at baseline from the original Dutch cohort [24]. A number of other studies have shown that young people with GD have higher scores on the CBCL or YSR than same-age population peers, and that they are similar to young people referred to clinical services for a range of mental health problems [44–46]. Population-based studies in America support higher baseline levels of mental health problems amongst young people with GD, with the prevalence of self-harm notably higher than for male or female peers [47,48]. Young people in our study had baseline YSR scores 0.7–1.0 SD higher than norms for age in comparable countries [29,46].

We found no evidence of change in psychological function with GnRHa treatment as indicated by parent report (CBCL) or self-report (YSR) of overall problems, internalising or externalising problems or self-harm. This is in contrast to the Dutch study which reported improved psychological function across total problems, externalising and internalising scores for both CBCL and YSR and small improvements in CGAS [24]. It also contrasts with a previous study from the UK GIDS of change in psychological function with GnRHa treatment in 101 older adolescents with GD (beginning > 15.5 years) which reported moderate improvements in CGAS score over 12 months of GnRHa treatment [49]. CGAS scores in this previous study increased from 61 to 67 with GnRHa treatment, similar to those (63 at baseline, 66 at 24 months) in our study. Follow-up of the Kuper et al. cohort found non-significant changes in depression and anxiety scores in those (n = 25) who had only pubertal suppression treatment, although improvements were seen in the whole sample combining these with those receiving cross-sex hormones [17]. A second US cohort reported that in 23 young people who had received pubertal suppression (using GnRHa or anti-androgens in birth-registered males and either GnRHa or medroxyprogesterone in birth-registered females), there was a reduction in depression scores in birth-registered males but not females.

A recent large US survey found that those who received pubertal suppression in early or mid adolescence had lower odds of lifetime suicidal ideation when studied in adulthood compared with those who did not, regardless of whether they later received cross-sex hormones

and after adjustment for a range of confounding factors [50]. This implies an enduring benefit of pubertal suppression on psychological function, however the cross-sectional design and retrospective exposure classification means the findings require replication. Data are also available from other conditions in which GnRHa is used to suppress puberty during adolescence. A trial of GnRHa suppression of puberty during early adolescence in young people born small-for-gestational-age (SGA) who were also treated with human growth hormone (GH) reported that those treated with GnRHa had similar cognitive and psychological function in adult life to those treated only with GH [51].

The differences between our findings and the previous GIDS study re change in psychological function may relate simply to sample size. But why our findings differ from those of the Dutch study is unclear. They may relate to the timing of assessments; we assessed young people multiple times whereas in the Dutch study the second assessment was shortly before starting cross-sex hormone treatment. Alternatively, there may have been baseline differences in the two cohorts. Whilst some aspects of psychological function were similar, as noted above, the baseline CGAS scores were notably higher in the Dutch group (indicating better function). A previous international comparison study has found that young people aged 12–18 years with GD from the UK have higher scores indicating greater problems on the CBCL and YSR than those from the Netherlands, Belgium and Switzerland [52].

Psychological distress and self-harm are known to increase across early adolescence. Normative data show rising YSR total problems scores with age from age 11 to 16 years in non-clinical samples from a range of countries [29]. Self-harm rates in the general population in the UK and elsewhere increase markedly with age from early to mid-adolescence, being very low in 10 year olds and peaking around age 16–17 years [53–56]. Our finding that psychological function and self-harm did not change significantly during the study is consistent with two main alternative explanations. The first is that there was no change, and that GnRHa treatment brought no measurable benefit nor harm to psychological function in these young people with GD. This is consonant with the action of GnRHa, which only stops further pubertal development and does not change the body to be more congruent with a young person's gender identity. The second possibility is that the lack of change in an outcome that normally worsens in early adolescence may reflect a beneficial change in trajectory for that outcome, i.e. that GnRHa treatment reduced this normative worsening of problems. In the absence of a control group, we cannot distinguish between these possibilities. We aimed to use normative reference data to examine this issue. However age- and gender-standardised t-scores for ASEBA and other outcomes cannot answer this question as they cover a very broad age range (e.g. 12–18 years). We had anticipated that z-scores on the YSR available by calendar year for two comparable countries (Netherlands; Australia) might be informative however confidence intervals were too wide to draw reliable inferences.

Gender dysphoria and body image changed little across the study. This is consistent with some previous reports [24] and was anticipated, given that GnRHa does not change the body in the desired direction, but only temporarily prevents further masculinization or feminization. Other studies suggest that changes in body image or satisfaction in GD are largely confined to gender affirming treatments such as cross-sex hormones or surgery [57]. We found that birth-registered sex and baseline pubertal status were not associated with later psychological functioning on GnRHa, consistent with previous reports [24,49].

These data correct reports from a recent letter by Biggs [58] which used preliminary data from our study which were uncleaned and incomplete data used for internal reporting. In addition there were many statistical comparisons which inflated the risk of type 1 error. Our statistical analysis plan restricted testing all outcomes for differences by sex due to the type 1

error risk. Contrary to Biggs's letter, we found no evidence of reductions over time in any psychological outcomes, and no material differences by sex.

## Strengths and limitations

Our study provides comprehensive data on this cohort during follow-up, with an anonymised dataset containing standardised scores deposited to allow other researchers to replicate our findings where data-sharing allows. The study size and uncontrolled design were key limitations. The small sample size limited our ability to identify small changes in outcomes. This was an uncontrolled observational study and thus cannot infer causality. Further, many of the outcomes studied here, including psychological function, self-harm and BMD, undergo normative changes by age and developmental stage during puberty that could confound any observed effect of GnRHa treatment in an uncontrolled study. The analysis plan aimed to take these issues into account as far as possible, however this particularly limits the potential for the study to show benefits or harms from treatment. However, some conclusions can be drawn. It is unlikely that the reported adverse events such as headaches do not relate directly to GnRHa treatment. Equally, given that there were no changes in psychological function and differences in point estimates were minimal for nearly all outcomes, it is unlikely that the treatment resulted in psychological harm. Observational studies are important sources of data on harms of treatment [59–61].

Our data are subject to a number of other limitations. This was an unfunded study undertaken within a clinical service and we were dependent on the clinical service for data collection. There were varying sample sizes for differing tests as some participants did not attend certain investigations and some follow-up medical tests were processed locally to patients; these data are reported as normal or otherwise. Missing items on psychological questionnaires resulted in some unusable data. Some young people found repeated completion of questionnaires about gender issues intrusive and refused to complete them at later follow-ups, as has been reported in other studies [62]. This questionnaire fatigue also affected parent responses. Scoring of psychological questionnaire data was rechecked at the completion of the study however this was not possible in very small numbers of participants in whom only scale scores rather than individual item data were preserved during data migration in hospital clinical information systems. In sensitivity analyses, repeat analysis of ASEBA psychological outcomes restricted to those with rescored data showed highly similar findings to the full sample (see S3 Table in S1 Appendix).

A more detailed qualitative evaluation of participant experience was not possible due to lack of interviewer time, and reporting of interview data was restricted to perceptions of positive or negative change and the giving of examples.

## Implications and conclusions

Treatment of young people with persistent and severe GD aged 12–15 years with GnRHa was efficacious in suppressing pubertal progression. Anticipated effects of withdrawal of sex hormones on symptoms were common and there were no unexpected adverse events. BMD increased with treatment in the lumbar spine and was stable at the hip, and BMD z-score fell consistent with delay of puberty. Overall participant experience of changes on GnRHa treatment was positive. We identified no changes in psychological function, quality of life or degree of gender dysphoria.

The great majority of this cohort went on to start cross-sex hormones, as was hypothesized given the severity and continuation of their GD. However one young person did not, providing some evidence that development of gender identity continues on GnRHa treatment and

confirming the importance of continuing supportive psychological therapy to allow further exploration of gender identity and a range of future pathways whilst on GnRHa.

This cohort will be followed up longer term to examine physical and mental health outcomes into early adulthood. However larger and longer-term prospective studies using a range of designs are needed to more fully quantify the harms and benefits of pubertal suppression in GD and better understand factors influencing outcomes [3]. These are beginning to be funded in a number of countries [63].(https://logicstudy.uk) Given that pubertal suppression may be both a treatment in its own right and also an intermediate step in a longer treatment pathway, it is essential for such studies to examine benefits and harms across the longer pathway including pubertal suppression and initiation of cross-sex hormones.

## Supporting information

**S1 Appendix.**
(DOCX)

**S2 Appendix. Statistical analysis plan.**
(DOCX)

## Acknowledgments

We wish to thank the young people and families who participated in the study and the clinical teams at The Tavistock and Portman NHS Foundation Trust and UCL Hospitals NHS Foundation Trust.

We wish to acknowledge the inputs of Harriet Gunn, Claudia Zitz and Domenico di Ceglie for their work in formulating the study, collecting data and advising on the manuscript.

## Author Contributions

**Conceptualization:** Polly Carmichael, Gary Butler, Elin M. Skageberg, Sophie Khadr, Russell M. Viner.

**Data curation:** Una Masic, Russell M. Viner.

**Formal analysis:** Tim J. Cole, Bianca L. De Stavola, Russell M. Viner.

**Investigation:** Gary Butler, Una Masic.

**Methodology:** Polly Carmichael, Bianca L. De Stavola, Elin M. Skageberg, Russell M. Viner.

**Writing – original draft:** Polly Carmichael, Gary Butler, Russell M. Viner.

**Writing – review & editing:** Polly Carmichael, Gary Butler, Una Masic, Tim J. Cole, Sarah Davidson, Sophie Khadr, Russell M. Viner.

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
