## [Decision Letter · Decision Letter 0]

22 Apr 2020

PONE-D-20-03178

Short-term outcomes of pubertal suppression in a selected cohort of 12 to 15 year old young people with persistent gender dysphoria in the UK

PLOS ONE

Dear Prof. Viner,

Thank you for submitting your manuscript to PLOS ONE. After careful consideration, we feel that it has merit but does not fully meet PLOS ONE’s publication criteria as it currently stands. Therefore, we invite you to submit a revised version of the manuscript that addresses the points raised during the review process.

We would appreciate receiving your revised manuscript by May 14 2020 11:59PM. To enhance the reproducibility of your results, we recommend that if applicable you deposit your laboratory protocols in protocols.io, where a protocol can be assigned its own identifier (DOI) such that it can be cited independently in the future. For instructions see: http://journals.plos.org/plosone/s/submission-guidelines#loc-laboratory-protocols

We look forward to receiving your revised manuscript.

Kind regards,

Geilson Lima Santana, M.D., Ph.D.

Academic Editor

PLOS ONE

Journal Requirements:

Reviewers' comments:

Reviewer's Responses to Questions

**Comments to the Author**

1. Is the manuscript technically sound, and do the data support the conclusions?

Reviewer #1: Partly

Reviewer #2: Yes

2. Has the statistical analysis been performed appropriately and rigorously? 

Reviewer #1: Yes

Reviewer #2: Yes

3. Have the authors made all data underlying the findings in their manuscript fully available?

Reviewer #1: No

Reviewer #2: No

4. Is the manuscript presented in an intelligible fashion and written in standard English?

Reviewer #1: Yes

Reviewer #2: Yes

5. Review Comments to the Author

Reviewer #1: This is a prospective observational uncontrolled study of a cohort of 44 adolescents with gender dysphoria treated with GnRH analogues, reporting both physical and psychological outcomes. The lack of a control group is an important limitation which the authors acknowledge and discuss. The use of z-scores for various measures does allow some comparison with expected changes with age.

The sample size is small (n=44), and even smaller for data beyond 12 months of follow-up, but given the fact that very limited outcome data of GnRHa treatment in adolescents with gender dysphoria are available, the study is a valuable addition nonetheless. However, the authors do need to clarify if any data from this cohort have been included in previous reports, for example in the studies by Costa et al. (ref 39) and by Joseph et al. (ref 48).

The authors provide a clear and detailed description of the methodology. Because of the small sample size the statistical analysis was limited to a small set of key outcomes that had been pre-specified in an analysis plan provided as supplemental material. Data are not being made available because of concerns about disclosure. The manuscript generally reads well.

Of the three aims of the study described in lines 129-134 the third receives relatively little attention.

I have the following questions and comments:

Introduction

Line 105: >20 years of follow-up is available on a single case only (ref 12) rather than on a cohort.

Methods

Lines 178-179: Why was pubic hair stage >2/3 a requirement in addition to breast/genital stage >2/3? (this is not the case in the Endocrine Society guideline for example)

Line 247: were options to store gametes available to all, for example, were birth-registered females aged <16 years eligible for cryopreservation of oocytes?

Line 251: was consent from one parent sufficient even if both parents held parental responsibility?

Line 281: ‘based predominantly on the breast/genital stage’ is unclear.

Were Tanner stages evaluated by the clinician or self-reported?

Lines 286-290: was a short synacthen test performed in all birth-assigned females, irrespective of symptoms/baseline adrenal steroid levels? Routinely performing a synacthen test and pelvic ultrasound is not recommended by the Endocrine Society guideline.

Lines 475: please clarify that ‘sexuality’ refers to sexual orientation rather than sexual activity

Results

Line 500: The median time between intake and study entry is 2.0 years. Does the diagnostic process typically take this long? Or did many adolescents have psychiatric disorders, for example autism spectrum disorder, which complicated the diagnostic assessment? This is also relevant in relation to the psychological outcomes.

Lines 506-507: most clinics report seeing more birth registered females than males in recent years. What might explain this difference in sex ratio with other clinics?

Lines 545-546: was the attempt to store sperm successful? Did it take 9 months for sperm production to resume?

Lines 563: the difference in lumbar spine BMD z-score does seem to be larger between 36 months and baseline (-1.5 - -0.2= -1.3) than between 24 months and baseline (-1.5 - -0.5= -1.0) which suggests they the z-score may further decrease between 24 and 36 months. The same is true for the height-adjust z-scores for the spine.

Line 594: The group at 24 months is not the same as the group at baseline. Were the scores within the subgroup with data at 24 months also higher at 24 months compared to baseline?

Lines 652-654: but one person eventually decided to stop GnRHa treatment. Could the authors describe the reasons for this decision? Did this person no longer experience gender dysphoria (this is suggested later on in line 889) or were there other reasons to decide against cross-sex hormone treatment?

Lines 662-663: could the authors provide some more detail on the add-back oestradiol therapy. What dose was prescribed, for what period and how effective was this in reducing symptoms?

Table 3. Why is the n different for BMC (42) and BMD (43) at the lumbar spine at 12 months follow-up?

Table 6. It would be helpful to mention in the table or legend that BMD is BMD at the lumber spine.

Table 7. Why are headaches and hot flushes reported together rather than separately?

Discussion

Lines 698-700: could the authors look at baseline BMI z-score of the 14 individuals from whom data are available at 36 months to assess which of the two explanations offered is more likely (true trend towards greater adiposity or effect of different subgroup at 36 months compared to baseline)?

Lines 714-715: why were side effects reported during interviews not included in table 7? (percentage fatigue at 13-24 months in the table is much lower (8%) than the percentage mentioned here (38%)).

Lines 715-717: headaches are very common among adolescents too so isn’t it difficult to conclude that they are due to GnRHa treatment, just as it is for fatigue? For example, a Norwegian study reports that the one-year prevalence of any headache type was 88% among 493 adolescents aged 12-18 years, and the point prevalence 38% (Cephalalgia. 2015 Nov;35(13):1181-91. doi: 10.1177/0333102415573512).

Line 813: ‘consistent with the fall in height z-score’ suggests that the fall in BMD z-score can be explained by the reduced growth rate but height-adjust z-scores also decreased.

Lines 850-851: see comments above about the headaches (lines 715-717)

Lines 851-853: perhaps the authors could add something about the potential influence of the treatment on gender identity development. In lines 888-892 they indicate that the fact that one person did not start cross-sex hormones provides evidence that gender identity can change during treatment but it is unknown if this number would have been different if the adolescents had not been treated with GnRHa. One of the aims of the study was to assess the persistence and desistence of GD so this could be discussed in some more detail.

In lines 194-204 the authors describe low baseline BMD as an exclusion criterion for treatment, although from lines 257-260 it seems that no one was actually excluded because of this. Do the authors still consider low BMD z-scores a contra-indication for GnRHa treatment based on the findings of the current study?

References

Reference 9 – author line is incorrect

Comments relating to language

Line 77: please remove parentheses

Line 164: change ‘stresses medical treatment’ into ‘stresses associated with medical treatment’

Lin 246: change ‘was’ to ‘were’

Reviewer #2: General;

This manuscript on the ‘early intervention study’ of the UK Gender Identity Development Service is long awaited and the results are extremely topical. At present, despite the fact that blockers are widely used in increasing numbers of transgender adolescents and recommended according to the Endocrine’s Society and WPATH’s guidelines, a replication of the first longitudinal Dutch study on the outcomes of this intervention is lacking. The current manuscript analyses Psychological Functioning (PF) and Body Image / Gender Dysphoria (GD) in addition to Physical outcomes after 12, 24 (and 36) months of the first 44 adolescents with gender dysphoria receiving blockers in the UK. The manuscript extensively describes the background, aims, methods and results of the study concluded by a detailed discussion of the findings. It refers to much relevant literature. It is sometimes too lengthly and not enough to the point. Below some comments and suggestions for further improvement and clarification.

Title; I think a title like ‘Physical, Psychological And Body Image Outcomes Of Pubertal Suppression In 12 To 15 Years Old Young People With Persistent Gender Dysphoria In The UK’ better represents the study

For clarity, in every part of the study (intro, methods, results and discussion) the order should be the same; first physical (including side effects), than psychological, followed by body image (and gender dysphoria). And then follow the same order of sub-subjects again within these main subjects.

Abstract:

- ‘semistructured interviews’ ; this part of the study is least explained and vague, also in the complete manuscript. Asks for some further explanation

- Tanner stages could be added here already

- ‘most participants reported positive or mixed positive-and-negative life changes’ unclear what this means, looking at the figure representing the outcomes, interview results seem more positive

- Conclusions; physical outcomes are missing, the results put in international perspective and a concluding message for clinicians is missing

Introduction:

- Introduction is rather detailed, especially about the UK situation and the Tavistock clinic, could be more general and shorter

- Terminology; persistence and desistence are introduced; these terms are under debate, especially since they are in the literature primarily used regarding the persistence or desistence of pre-pubertal GD; I suggest to stick to the ‘continuing or discontinued wish for puberty suppression’

- On p 3, when describing the guidelines, the criteria for puberty suppression should be added, like they are described in the SOC’s and the Endocrine guidelines, these are extensively described

- P. 4 when referring to a meta-analysis of over 500 young people, what were the conclusions in the review?

- Long term follow up data, again, what are the conclusions or outcomes of the referred papers? The 20 year plus follow up is just one case.

- Aims; change persistence and desistence in ‘continuation’ and ‘discontination’

- P. 5, last paragraph of the Introduction, ‘many of the … to assess the effects of treatment’; this is a limitation that better belongs in the Discussion

- Tables; add ‘sex’ to primary characteristics, secondary characteristics

Methods

- Methods are lengthy described and a bit diffuse, could be more to the point and shorter

- Participants; recruited from clinical cohorts…; which cohorts are referred to?

- ‘potentially eligible’ ; when was an adolescent considered ‘potentially eligible’? what were eligibility criteria?

- ‘fulfils standard readiness criteria’ ; what were these and weren’t these the same as the eligibility criteria?

- It might be helpful to provide the eligibility criteria of the SOC and how these were adapted in the GIDS, e.g. the SOC do not ask for ‘6 months of and at least 4 interviews assessment’; it are four criteria (persisting and intensifying GD, no interfering mental health comorbities, family / social support, capacity for informed consent), in addition to Tanner stage 2-3 (and age 12 in the case of the Dutch clinic where suppression was introduced, de Vries 2012, etc)

- P 6; exclusion criteria; how many adolescents were excluded? Due to what reason? Exclusion criteria overlap with ‘not fulfilling eligibility criteria’ (e.g interfering comorbidties; an example would be needle phobia). BMI < 2nd centile, what about BMI in higher range/ overweight?, 5 ii, iii, and iv; ‘able to give informed consent’

- P 8, first para; how many families were no longer interested 3 months after the first visit?

- P.8 third para; how many adolescents stored gametes?

- P. 8 fifth para; ‘a small number’ ; how many?

- P. 8 last para; participants were ‘followed up’; how?

- P. 10; when were questionnaires collected? What was the procedure?

- P. 11: ‘self harm’, change into ‘suicidal thoughts and self-harm’

- P 12; 5 participant experience and satisfaction

o This is an interesting part, what sort of questions were asked and how were they analysed?

- P 13, third para; ‘note that……a future paper’ doesn’t need to be here, can be deleted or part of the discussion

Results

- 7 tables and 1 figure is quite extensive, maybe some findings could be presented in a more compact way, Tables can maybe combined? Is the figure necessary? Could also be presented in text

- P 20, the description of the individuals temporarily stopping blockers is very interesting; was the sperm storage successful?

- Table 4; internalizing and externalizing scores are also provided but not discussed in the methods and discussion?

Discussion

- The discussion is very extensive and detailed with a lot of repetition of the results, could be more compact and to the point

- Third para; …or reflect a higher baseline in this group; baseline BMI can be compared with norm reference values?

- Finding of experiencing ‘fatigue or low energy’ as side effects that are unknown in GnRHanalogues deserve more discussion, might these be subjective attributions of the adolescents?

- ‘comparison with the literature’ delete this heading, a discussion is always an comparison with the literature

- After strengths and limitations, a concluding paragraph with clinical implications should be added as a final section

6. PLOS authors have the option to publish the peer review history of their article (what does this mean?). If published, this will include your full peer review and any attached files.

Reviewer #1: No

Reviewer #2: No

---

## [Author Response · Author response to Decision Letter 0]

9 Jul 2020

Dear Editor

Many thanks for asking us to revise and resubmit this paper. 

We thank the reviewers for their thorough evaluations and helpful comments and respond to issues point by point below. 

Reviewer #1

1. the authors do need to clarify if any data from this cohort have been included in previous reports, for example in the studies by Costa et al. (ref 39) and by Joseph et al. (ref 48).

Response: We believe there is no overlap with patients in Costa et al as that group were substantially older when they commenced GnRHa. There is some overlap in patients with Joseph et al.; all 44 patients in our current paper were included and together others in the clinic from 2011-2016 so our patient group is included within the larger cohort of 70 reported in the Joseph et al paper. Note that paper specifically examined bone mineral apparent density (BMAD), an alternative parameter to those reported here. Different methods of analysis of the data show the same trend.

2. Introduction. Line 105: >20 years of follow-up is available on a single case only (ref 12) rather than on a cohort.

Response: We have changed this to “Longer-term follow-up data are limited to individuals from one cohort” and included a number of references including the case-report on the single case. 

3. Methods Lines 178-179: Why was pubic hair stage >2/3 a requirement in addition to breast/genital stage >2/3? (this is not the case in the Endocrine Society guideline for example)

Response: These were the criteria set out at the time and we recognise they did not entirely match Endocrine Society guidance. We could remove the PH data if this is felt appropriate.

4. Line 247: were options to store gametes available to all, for example, were birth-registered females aged <16 years eligible for cryopreservation of oocytes?

Response: No. All patients were offered counselling and referral for available services. The offer of cryopreservation depended on what was available from the National Health Service at the time. The patients were seen from April 2011, a time when cryopreservation was not routinely available. We have noted in the revised paper that “Access to storage depended on regional availability within the NHS.”

5. Line 251: was consent from one parent sufficient even if both parents held parental responsibility?

Response: Yes, under English law. 

6. Line 281: ‘based predominantly on the breast/genital stage’ is unclear. Were Tanner stages evaluated by the clinician or self-reported?

Response: Tanner stage was evaluated by a clinician – this is now noted Line 243. Line 281 is now clarified: “This was summarized into a single pubertal stage, with the breast/genital stage taking precedence if there was a discrepancy between breast/genital and pubic hair stage.” As noted above, we would be happy to remove the data on PH stage if recommended; this would make no difference as B/G stage was used for the summary if any discrepancy.

7. Lines 286-290: was a short synacthen test performed in all birth-assigned females, irrespective of symptoms/baseline adrenal steroid levels? Routinely performing a synacthen test and pelvic ultrasound is not recommended by the Endocrine Society guideline.

Response: Yes. It is important to note that the study was planned in 2008-10 at a time when it was routine to undertake short synacthen tests. Some of the investigations went beyond Endocrine Society recommendations as was appropriate for a consented study at the time. 

8. Lines 475: please clarify that ‘sexuality’ refers to sexual orientation rather than sexual activity

Response: Sexuality refers to a general sense of the young person’s sexual sense, broader than but including orientation. 

9. Results. Line 500: The median time between intake and study entry is 2.0 years. Does the diagnostic process typically take this long? Or did many adolescents have psychiatric disorders, for example autism spectrum disorder, which complicated the diagnostic assessment? This is also relevant in relation to the psychological outcomes.

Response: The length of assessment can vary depending on individual needs. This can be for a number of reasons including for example, psychological or mental health difficulties or lack of family support. However in this case the median time in the service between intake to the service and intake into the study reflects the age range of participants at the time of referral to the service. It could be a number of years before a young person reached the age of 12 years and was eligible for intake into the study. 

10. Lines 506-507: most clinics report seeing more birth registered females than males in recent years. What might explain this difference in sex ratio with other clinics?

Response: First this was a highly selected sample and thus it would be plausible to see a different sex ratio in this group compared with a referred clinical sample. Second, recruitment occurred 2011-14. 

11. Lines 545-546: was the attempt to store sperm successful? Did it take 9 months for sperm production to resume?

Response: Apologies however we believe it is not appropriate to include this information. First, the outcomes of gamete storage are outside the remit of this study as they were not included in our protocol and we currently do not access to these data for all patients. This is because fertility assessments/treatments were conducted by the National Health Service locally to the patient and access to these data was not part of our research protocol. Second, providing highly sensitive data on a single patient may be an identification risk in such a small cohort. 

12. Lines 563: the difference in lumbar spine BMD z-score does seem to be larger between 36 months and baseline (-1.5 - -0.2= -1.3) than between 24 months and baseline (-1.5 - -0.5= -1.0) which suggests they the z-score may further decrease between 24 and 36 months. The same is true for the height-adjust z-scores for the spine.

Response: We believe that there are insufficient data to comment on this as confidence intervals are very wide. The group at 36 months is different to, and half of the size of, the group at 24 months. 

13. Line 594: The group at 24 months is not the same as the group at baseline. Were the scores within the subgroup with data at 24 months also higher at 24 months compared to baseline?

Response: We believe this relates to the testing of differences between baseline and 24 months. Testing using paired t-tests i.e. restricted to those with data at both baseline and 24 months. 

14. Lines 652-654: but one person eventually decided to stop GnRHa treatment. Could the authors describe the reasons for this decision? Did this person no longer experience gender dysphoria (this is suggested later on in line 889) or were there other reasons to decide against cross-sex hormone treatment?

Response: Young people continue with therapeutic work alongside GnRHa treatment. The young person had continued uncertainty about taking cross-sex hormones, which included concerns about possible side effects. We have briefly noted this in the revised paper although again we feel it is not appropriate to include detailed data on individuals. 

15. Lines 662-663: could the authors provide some more detail on the add-back oestradiol therapy. What dose was prescribed, for what period and how effective was this in reducing symptoms?

Response: Add-back oestradiol was suggested at 5 micrograms of ethinyl oestradiol however this was prescribed by general practitioners and not directly under the control of the clinicians. This was effective at reducing symptoms in each case. This has been added to the revised manuscript. 

16. Table 3. Why is the n different for BMC (42) and BMD (43) at the lumbar spine at 12 months follow-up?

Response: This is because only the BMD data were available for this patient at 12 months not the BMC data. 

17. Table 6. It would be helpful to mention in the table or legend that BMD is BMD at the lumbar spine.

Response: Done.

18. Table 7. Why are headaches and hot flushes reported together rather than separately?

Response: For two reasons; first they likely share a common aetiology in terms of hormone withdrawal; and second these symptoms were frequently about asked together by clinicians. 

Discussion

19. Lines 698-700: could the authors look at baseline BMI z-score of the 14 individuals from whom data are available at 36 months to assess which of the two explanations offered is more likely (true trend towards greater adiposity or effect of different subgroup at 36 months compared to baseline)?

Response: We took a decision to limit the amount of analysis in this very small cohort and feel that to undertake this testing would add to the risk of chance associations being identified. 

20. Lines 714-715: why were side effects reported during interviews not included in table 7? (percentage fatigue at 13-24 months in the table is much lower (8%) than the percentage mentioned here (38%)).

Response: We reported potential adverse events identified in clinics in Table 7. These were issues identified through clinician assessment – and here this was 8%. We believe that these are different to perceptions of change in energy levels reported from the semi-structured interviews (i.e. the 38%). One represents a clinical report of fatigue whereas the other represents a response that the young person has perceived some reduction in energy levels although the amount is not quantified.

21. Lines 715-717: headaches are very common among adolescents too so isn’t it difficult to conclude that they are due to GnRHa treatment, just as it is for fatigue? For example, a Norwegian study reports that the one-year prevalence of any headache type was 88% among 493 adolescents aged 12-18 years, and the point prevalence 38% (Cephalalgia. 2015 Nov;35(13):1181-91. doi: 10.1177/0333102415573512).

Response: Agreed. We were taking a conservative point of view that headaches might be due to the treatment. We note this now in the revised manuscript, quoting this helpful reference: “The relationship of symptoms such as headaches, fatigue and sleep disturbance to GnRHa treatment is unclear as each of these are very common symptoms amongst early adolescents,[32, 33] although a conservative perspective would regard them as side-effects of treatment.”

22. Line 813: ‘consistent with the fall in height z-score’ suggests that the fall in BMD z-score can be explained by the reduced growth rate but height-adjusted z-scores also decreased.

Response: We noted that the fall in BMD z-score was consistent with the same observed fall in height z-score, as both likely occurred because the young people had much reduced growth. As the reviewer noted, height-adjusted BMD z-scores also fell. This may reflect that the fall in BMD z-score was higher than the fall in height z-score – or that height-adjustment of BMI z-score is imperfect. We have reworded this as follows: “Our findings that BMD increased over time in the lumbar spine but more slowly than in same age peers resulting in a fall in z-score, are similar to findings from a larger study from the UK[49] and to international studies.[2, 12, 50] The fall in height-adjusted BMD z-score was consistent with but larger than the fall in height z-score.”

23. Lines 850-851: see comments above about the headaches (lines 715-717)

Response: See response 21 above re headaches. 

24. Lines 851-853: perhaps the authors could add something about the potential influence of the treatment on gender identity development. In lines 888-892 they indicate that the fact that one person did not start cross-sex hormones provides evidence that gender identity can change during treatment but it is unknown if this number would have been different if the adolescents had not been treated with GnRHa. One of the aims of the study was to assess the persistence and desistence of GD so this could be discussed in some more detail.

Response: See response to point 14 above. 

25. In lines 194-204 the authors describe low baseline BMD as an exclusion criterion for treatment, although from lines 257-260 it seems that no one was actually excluded because of this. Do the authors still consider low BMD z-scores a contra-indication for GnRHa treatment based on the findings of the current study?

Response: As outlined in our exclusion criteria, a low BMD was considered acceptable if certain criteria were met. No patients were excluded on this basis. In terms of the question about low BMD z-scores being a contra-indication, our study design does not allow us to confidently answer this question. 

References

26. Reference 9 – author line is incorrect

Response: Corrected.

27. Comments relating to language

Line 77: please remove parentheses

Line 164: change ‘stresses medical treatment’ into ‘stresses associated with medical treatment’

Lin 246: change ‘was’ to ‘were’

Response: Done.

Reviewer #2: 

1. Title; I think a title like ‘Physical, Psychological And Body Image Outcomes Of Pubertal Suppression In 12 To 15 Years Old Young People With Persistent Gender Dysphoria In The UK’ better represents the study

Response: We believe that the current title is appropriate, particularly noting it provides only short-term outcomes and is applicable to a highly selected cohort. 

2. For clarity, in every part of the study (intro, methods, results and discussion) the order should be the same; first physical (including side effects), than psychological, followed by body image (and gender dysphoria). And then follow the same order of sub-subjects again within these main subjects.

Response: The Methods and Result sections largely take this order, although we believe it is most appropriate to discuss side effects and qualitative findings last. We have further edited the paper to shorten where possible and taken into account this suggestion where appropriate. In the Discussion section we discuss the main findings first and then compare findings with the literature, but within these sections we have used this order. 

Abstract:

3. ‘semistructured interviews’ ; this part of the study is least explained and vague, also in the complete manuscript. Asks for some further explanation

Response: Apologies there is no room for additional detail in the abstract. We have put additional detail into the Methods section although have been mindful of the already high word count. 

4. Tanner stages could be added here already

Response: We believe that these data are too detailed for the abstract word count. 

5. ‘most participants reported positive or mixed positive-and-negative life changes’ unclear what this means, looking at the figure representing the outcomes, interview results seem more positive

Response: We have reworded this as follows: “Most participants reported positive or a mixture of positive and negative life changes on GnRHa.”

6. Conclusions; physical outcomes are missing, the results put in international perspective and a concluding message for clinicians is missing

Response: We added the sentence “Changes in BMD were consistent with suppression of growth.” Apologies however we do not believe that the complexity of messages for clinicians can be summarized in the abstract. 

Introduction:

7. Introduction is rather detailed, especially about the UK situation and the Tavistock clinic, could be more general and shorter

Response: We have re-edited the paper to shorten it where possible.

8. Terminology; persistence and desistence are introduced; these terms are under debate, especially since they are in the literature primarily used regarding the persistence or desistence of pre-pubertal GD; I suggest to stick to the ‘continuing or discontinued wish for puberty suppression’

Response: We agree and have amended these in the revised paper.

9. On p 3, when describing the guidelines, the criteria for puberty suppression should be added, like they are described in the SOC’s and the Endocrine guidelines, these are extensively described

Response: The criteria used in this study are described in the Methods section. 

10. P. 4 when referring to a meta-analysis of over 500 young people, what were the conclusions in the review?

Response: The findings from this review are discussed in the various sections in the Discussion. 

11. Long term follow up data, again, what are the conclusions or outcomes of the referred papers? The 20 year plus follow up is just one case.

Response: Again the findings of these studies are outlined in the Discussion. Re the 20 year follow-up we have reworded this as suggested by Reviewer 1.

12. Aims; change persistence and desistence in ‘continuation’ and ‘discontination’

Response: See response to point 8.

13. P. 5, last paragraph of the Introduction, ‘many of the … to assess the effects of treatment’; this is a limitation that better belongs in the Discussion

Response: We have moved this paragraph to the Discussion as suggested. 

14. Tables; add ‘sex’ to primary characteristics, secondary characteristics

Response: Apologies we did not understand this request. 

Methods:

15. Methods are lengthy described and a bit diffuse, could be more to the point and shorter

Response: We have shortened the paper where possible.

16. Participants; recruited from clinical cohorts…; which cohorts are referred to?

Response: These were patients referred for specialist treatment. We have reworded this to read “They were all recruited from patients referred to the GIDS.”

17. ‘potentially eligible’ ; when was an adolescent considered ‘potentially eligible’? what were eligibility criteria?

Response: We have removed this sentence – as we then define eligibility criteria and procedures which make it clear that those judged potentially eligible by the psychological service were referred to the endocrine liaison clinic for full assessment of eligibility.

18. ‘fulfils standard readiness criteria’ ; what were these and weren’t these the same as the eligibility criteria?

Response: These were explicitly part of the eligibility criteria and are defined in the following sentence (“relating to psychological stability sufficient to withstand the stresses of medical treatment for GID.”)

19. It might be helpful to provide the eligibility criteria of the SOC and how these were adapted in the GIDS, e.g. the SOC do not ask for ‘6 months of and at least 4 interviews assessment’; it are four criteria (persisting and intensifying GD, no interfering mental health comorbities, family / social support, capacity for informed consent), in addition to Tanner stage 2-3 (and age 12 in the case of the Dutch clinic where suppression was introduced, de Vries 2012, etc)

Response: This paper only relates to the eligibility criteria used in this study and we believe it would be confusing to list other eligibility criteria which were not directly applied to this study. That said, our eligibility criteria were essentially the same as in the Dutch clinic. We list them here in a slightly different format as these were the format used in our protocol. 

20. P 6; exclusion criteria; how many adolescents were excluded? Due to what reason? Exclusion criteria overlap with ‘not fulfilling eligibility criteria’ (e.g interfering comorbidties; an example would be needle phobia). BMI < 2nd centile, what about BMI in higher range/ overweight?, 5 ii, iii, and iv; ‘able to give informed consent’

Response: No young people were excluded after assessment, as only those meeting the psychological and consent criteria were counselled about the study and referred by GIDS to the endocrine clinic for consideration. We did not exclude any young people on the basis of high BMI as this was not one of our criteria. As noted in the 4th paragraph of p8, a small number of young people were not eligible for participation at the second medical visit as they were not yet sufficiently advanced in puberty. One of this group also had low BMI initially however this subsequently rose and was above the BMI threshold when the young person met puberty criteria after a number of reviews. 

21. P 8, first para; how many families were no longer interested 3 months after the first visit?

Response: We note in the 4th paragraph of p8 that 4 families did not wish to continue after the first/second visits. 

22. P.8 third para; how many adolescents stored gametes?

Response: As noted previously, the outcomes of gamete storage are outside the remit of this study as they were not included in our protocol. Additionally fertility assessments/treatments were conducted by the National Health Service locally to the patient and we do not currently have access to those data. 

23. P. 8 fifth para; ‘a small number’ ; how many?

Response: Thank you – it was helpful to go back and summarize this. There were 8 subjects and this has been added to the revised paper as follows: “Eight young people (7 birth assigned males) were not eligible for participation at the second medical visit as they were not yet sufficiently advanced in puberty. They were followed-up every 3-6 months and entered the study subsequently when sufficiently advanced in puberty (median waiting time 7 months).” 

24. P. 8 last para; participants were ‘followed up’; how?

Response: We have added “in the endocrine clinic” to explain they were followed up in the endocrine clinic. 

25. P. 10; when were questionnaires collected? What was the procedure?

Response: This is explained on p10 as follows: “A standardised set of psychological questionnaires used in the GIDS clinic was completed at the time young people were deemed potentially eligible and referred to the medical clinic. Questionnaires were completed at home by the young person and parent between GIDS clinical meetings. A research assistant followed up families to ensure completion of questionnaires. Questionnaires were repeated approximately every 12 months on treatment.”

26. P. 11: ‘self harm’, change into ‘suicidal thoughts and self-harm’

Response: We have changed this to read “Self-harm actions and thoughts were assessed…”.

27. P 12; 5 participant experience and satisfaction

o This is an interesting part, what sort of questions were asked and how were they analysed?

Response: We believe this is already described on p12 i.e. for each of the 10 domains (life overall, memory, focus, sense of direction, mood, energy levels, relationships with friends, relationships with family, gender role and sexuality) young people were asked a) about direction of changes since starting GnRHa; b) examples of changes experienced; and c) why they assigned the chosen change rating. To help with clarity we have modified the description on p12. 

28 P 13, third para; ‘note that……a future paper’ doesn’t need to be here, can be deleted or part of the discussion

Response: Apologies however we believe it important to note this here as this scale was mentioned in our Protocol. 

Results:

29. 7 tables and 1 figure is quite extensive, maybe some findings could be presented in a more compact way, Tables can maybe combined? Is the figure necessary? Could also be presented in text

Response: We hope the editor will agree that in an online only paper it is reasonable to include these Figures and Tables as our aim is to communicate results as fully as possible. 

30. P 20, the description of the individuals temporarily stopping blockers is very interesting; was the sperm storage successful?

Response: See response to point 22 above. 

31. Table 4; internalizing and externalizing scores are also provided but not discussed in the methods and discussion?

Response: These scores were provided with 95% CI and not tested. We note that there was no evidence of change in ASEBA scores. 

Discussion:

32. The discussion is very extensive and detailed with a lot of repetition of the results, could be more compact and to the point

Response: We have edited the paper to shorten where possible. 

33. Third para; …or reflect a higher baseline in this group; baseline BMI can be compared with norm reference values?

Response: Here we refer to BMI z-scores which are adjusted for age and sex. 

34. Finding of experiencing ‘fatigue or low energy’ as side effects that are unknown in GnRH analogues deserve more discussion, might these be subjective attributions of the adolescents?

Response: Apologies we were not exactly certain of the intent of this comment. It is unclear whether the fatigue and low energy are related to treatment or reflect the fact that these symptoms are extremely common in young people, as we noted. 

35. ‘comparison with the literature’ delete this heading, a discussion is always an comparison with the literature

Response: Done.

36. After strengths and limitations, a concluding paragraph with clinical implications should be added as a final section

Response: Our paper concludes with an Implications and Conclusions section, which summarizes findings and the need for further research. We feel that the role of this paper is to present the data from this small uncontrolled cohort as fully as possible in as balanced a way as possible, but it is not to advise clinicians on how to treat their patients. 

Editor comments

Response: Done

Response: The data in this study are highly sensitive as they relate to gender identity, sexuality and psychological function. Additionally, the cohort is very small and all were treated in one clinic. Thus the potential for identifiability is very high. We have explored how we can place a dataset on public access that protects patient privacy and respects their rights. We have made progress in this regard and will be placing an anonymised dataset in the UK Data Service ReShare data archive (www.ukdataservice.ac.uk); initial permissions have been obtained from the UK Health Research Authority and outline permission from the Data Service to deposit the data as Data Collection #854413. We are currently finalising the anonymisation and disclosure controls and notifying participants of the deposit. This will go live soon after publication of the results. 

 Response: Done

---

## [Decision Letter · Decision Letter 1]

4 Aug 2020

PONE-D-20-03178R1

Short-term outcomes of pubertal suppression in a selected cohort of 12 to 15 year old young people with persistent gender dysphoria in the UK

PLOS ONE

Dear Dr. Viner,

Thank you for submitting your manuscript to PLOS ONE. After careful consideration, we feel that it has merit but does not fully meet PLOS ONE’s publication criteria as it currently stands. Therefore, we invite you to submit a revised version of the manuscript that addresses the points raised during the review process.

We look forward to receiving your revised manuscript.

Kind regards,

Geilson Lima Santana, M.D., Ph.D.

Academic Editor

PLOS ONE

Additional Editor Comments (if provided):

Thank you for addressing reviewer's considerations.

Some minor aspects still need to be addressed, though. Please, take into consideration the reviewer's orientations, and, please, follow PlosOne's Guideline about the three levels of heading. https://journals.plos.org/plosone/s/file?id=80c1/PLOSOne_formatting_sample_main_body.pdf

Reviewers' comments:

Reviewer's Responses to Questions

**Comments to the Author**

1. If the authors have adequately addressed your comments raised in a previous round of review and you feel that this manuscript is now acceptable for publication, you may indicate that here to bypass the “Comments to the Author” section, enter your conflict of interest statement in the “Confidential to Editor” section, and submit your "Accept" recommendation.

Reviewer #2: All comments have been addressed

2. Is the manuscript technically sound, and do the data support the conclusions?

Reviewer #2: Yes

3. Has the statistical analysis been performed appropriately and rigorously? 

Reviewer #2: Yes

4. Have the authors made all data underlying the findings in their manuscript fully available?

Reviewer #2: (No Response)

5. Is the manuscript presented in an intelligible fashion and written in standard English?

Reviewer #2: (No Response)

6. Review Comments to the Author

Reviewer #2: This revised manuscript addressed most of my previous reviewer comments and deserves publication. The collected data have long been waited for publication, and the findings deserve an open scientific discussion.

I have just some minor comments that should be addressed before publication. First, while revising the manuscript, two papers have been published that should be incorporated and referred to and discussed. Second, some suggestions to interpret the outcomes in a slighty different way.

1) The two studies that are published in between revising the manuscript and reviewing are:

- Kuper LE, Stewart S, Preston S, et al. Body Dissatisfaction and Mental Health Outcomes of Youth on Gender-Affirming Hormone Therapy. Pediatrics. 2020; 145(4):e20193006

This is another proper follow up study, the first apart from the Dutch studies referred to in the Introduction (line 95, references 14-16), it is an important paper that shows improvement of psychological functioning and body dissatisfaction, although it did not evaluate effects of puberty suppression only but combined with affirming sex hormones

- Biggs, M, Gender Dysphoria and Psychological Funtioning in Adolescents Treated with GnRHa: comparing Dutch and English Prospective Studies, Arch Sex Beh, 2020

- This is a letter to the editor presenting the preliminary presented results of the data of the current study in comparison with the published Dutch results; although a detailed discussion of this letter deserves a separate commentary, some sentences should be included in the Discussion or Introduction of the present paper.

2) The findings of the semi structured interviews showed that the majority reported improved mood, family, friendship and gender role quality, although a smaller minority had more mixed experiences. These qualitative findings received just one sentence in the Discussion (678-681), but seem to me probably the most important finding, given that the numbers were further possibly too small to show improvements in quantitative measures! These more positive (and a bit mixed) results deserve therefor more positive attention in the Discussion and conclusion.

7. PLOS authors have the option to publish the peer review history of their article (what does this mean?). If published, this will include your full peer review and any attached files.

Reviewer #2: No

---

## [Author Response · Author response to Decision Letter 1]

5 Nov 2020

Editor comments

please, follow PlosOne's Guideline about the three levels of heading. https://journals.plos.org/plosone/s/file?id=80c1/PLOSOne_formatting_sample_main_body.pdf

Response: Done

Reviewer #2: 

I have just some minor comments that should be addressed before publication. First, while revising the manuscript, two papers have been published that should be incorporated and referred to and discussed. Second, some suggestions to interpret the outcomes in a slighty different way.

1) The two studies that are published in between revising the manuscript and reviewing are:

- Kuper LE, Stewart S, Preston S, et al. Body Dissatisfaction and Mental Health Outcomes of Youth on Gender-Affirming Hormone Therapy. Pediatrics. 2020; 145(4):e20193006

This is another proper follow up study, the first apart from the Dutch studies referred to in the Introduction (line 95, references 14-16), it is an important paper that shows improvement of psychological functioning and body dissatisfaction, although it did not evaluate effects of puberty suppression only but combined with affirming sex hormones

- Biggs, M, Gender Dysphoria and Psychological Funtioning in Adolescents Treated with GnRHa: comparing Dutch and English Prospective Studies, Arch Sex Beh, 2020

- This is a letter to the editor presenting the preliminary presented results of the data of the current study in comparison with the published Dutch results; although a detailed discussion of this letter deserves a separate commentary, some sentences should be included in the Discussion or Introduction of the present paper.

Response: We thank the reviewer and have referenced the first paper as a longitudinal outcome study and discussed psychological findings, although as the reviewer says, only a small number had only GnRHa treatment. We also note an additional paper on a longitudinal US cohort, Achille et al. https://doi.org/10.1186/s13633-020-00078-2, which reported outcomes of 23 having pubertal suppression (using either GnRHa or medroxyprogesterone) over 3 years and an additional paper on a second Dutch cohort (from Leiden) which provides data on the numbers continuing from GnRHa to cross-sex hormones (Brik et al. https://doi.org/10.1007/s10508-020-01660-8) .

We now note in our Background that longer-term follow-up data are limited to individuals from four cohorts, and provided additional information in the Discussion as follows:

“Our finding that 1 participant ceased pubertal suppression and did not commence cross-sex hormones is somewhat similar to the experience of one Dutch and one US cohort; Kuper et al. described that 2 of approximately 57 young people aged 10-15 years who commenced pubertal suppression treatment stopped this treatment without commencing cross-sex hormones. Brik et al. reported that in a cohort of 137 young people who began GnRHa between 10 and 18 years and were followed until eligible to commence cross-sex hormones, 5 (3.6%) ceased treatment and did not later commence cross-sex hormones.” 

“Three longitudinal studies from the Netherlands and the USA have examined psychological function over time in cohorts of young people treated with GnRHa and then cross-sex hormones,[17, 18, 23] although the two US cohorts were of limited size.”

Follow-up of the Kuper et al. cohort found non-significant changes in depression and anxiety scores in those (n=25) who had only pubertal suppression treatment, although improvements were seen in the whole sample combining these with those receiving cross-sex hormones.[17] A second US cohort reported that in 23 young people who had received pubertal suppression (using GnRHa or anti-androgens in birth-registered males and either GnRHa or medroxyprogesterone in birth-registered females), there was a reduction in depression scores in birth-registered males but not females.”

The second paper (by Biggs) appears to use fragmentary preliminary data from this study that were used for internal reporting in 2015. These data were both uncleaned and incomplete. We note in our revised paper the following:

“These data correct reports from a recent letter by Biggs which used preliminary data from our study which were uncleaned and incomplete used for internal reporting. In addition there were many statistical comparisons which inflated the risk of type 1 error. Our statistical analysis plan restricted testing all outcomes for differences by sex due to the type 1 error risk. Contrary to Biggs’s letter, we found no evidence of reductions over time in any psychological outcomes, and no material differences by sex.”

2) The findings of the semi structured interviews showed that the majority reported improved mood, family, friendship and gender role quality, although a smaller minority had more mixed experiences. These qualitative findings received just one sentence in the Discussion (678-681), but seem to me probably the most important finding, given that the numbers were further possibly too small to show improvements in quantitative measures! These more positive (and a bit mixed)

results deserve therefor more positive attention in the Discussion and conclusion.

Response: We have expanded the discussion of the semi-structured interviews as suggested by the reviewer: 

“Participant experience of treatment as reported in interviews was positive for the majority, particularly relating to feeling happier, feeling more comfortable, better relationships with family and peers and positive changes in gender role. Smaller numbers reported having mixed positive and negative changes. A minority (12% at 6-15 months and 17% at 15-24 months) reported only negative changes, which were largely related to anticipated side effects. None wanted to stop treatment due to side effects or negative changes. We are not aware of comparative patient experience data from other cohorts.”

---

## [Decision Letter · Decision Letter 2]

1 Dec 2020

Short-term outcomes of pubertal suppression in a selected cohort of 12 to 15 year old young people with persistent gender dysphoria in the UK

PONE-D-20-03178R2

Dear Dr. Viner,

We’re pleased to inform you that your manuscript has been judged scientifically suitable for publication and will be formally accepted for publication once it meets all outstanding technical requirements.

Kind regards,

Geilson Lima Santana, M.D., Ph.D.

Academic Editor

PLOS ONE

Additional Editor Comments (optional):

Reviewers' comments:

Reviewer's Responses to Questions

**Comments to the Author**

1. If the authors have adequately addressed your comments raised in a previous round of review and you feel that this manuscript is now acceptable for publication, you may indicate that here to bypass the “Comments to the Author” section, enter your conflict of interest statement in the “Confidential to Editor” section, and submit your "Accept" recommendation.

Reviewer #2: All comments have been addressed

2. Is the manuscript technically sound, and do the data support the conclusions?

Reviewer #2: Yes

3. Has the statistical analysis been performed appropriately and rigorously? 

Reviewer #2: Yes

4. Have the authors made all data underlying the findings in their manuscript fully available?

Reviewer #2: Yes

5. Is the manuscript presented in an intelligible fashion and written in standard English?

Reviewer #2: Yes

6. Review Comments to the Author

Reviewer #2: Authors addressed all my comments on the first revision and even more, which makes the current version up to date and important and relevant for transgender care for adolescents.

7. PLOS authors have the option to publish the peer review history of their article (what does this mean?). If published, this will include your full peer review and any attached files.

Reviewer #2: No

---

## [Editor Report · Acceptance letter]

19 Jan 2021

PONE-D-20-03178R2 

Short-term outcomes of pubertal suppression in a selected cohort of 12 to 15 year old young people with persistent gender dysphoria in the UK 

Dear Dr. Viner:

I'm pleased to inform you that your manuscript has been deemed suitable for publication in PLOS ONE. Congratulations! Your manuscript is now with our production department. 

Kind regards, 

on behalf of

Dr. Geilson Lima Santana 

Academic Editor

PLOS ONE